

Atmospheric
Chemistry
and Physics



# Seasonal impact of biogenic very ~~short lived~~ bromine on lowermost stratospheric ozone between 60° N and 60° S during the 21st century

**Javier Alejandro Barrera**[1], **Rafael Pedro Fernandez**[1,2,3], **Fernando Iglesias-Suarez**[2], **Carlos Alberto Cuevas**[2], **Jean-Francois Lamarque**[4], and **Alfonso Saiz-Lopez**[2]

[1]Institute for Interdisciplinary Science, National Research Council (ICB-CONICET),
FCEN-UNCuyo, Mendoza, 5500, Argentina
[2]Department of Atmospheric Chemistry and Climate, Institute of Physical Chemistry Rocasolano,
CSIC, Madrid, 28006, Spain
[3]Atmospheric and Environmental Studies Group (GEAA), UTN-FRM, Mendoza, 5500, Argentina
[4]Atmospheric Chemistry, Observations & Modelling Laboratory, National Center for Atmospheric Research,
Boulder, CO 80301, USA

**Correspondence:** Alfonso Saiz-Lopez (a.saiz@csic.es) and Rafael Pedro Fernandez (~~rpfernandez@conicet.gov.ar~~)

**Abstract.** [TS1]Biogenic very short lived ~~bromine~~ (VSL$^{Br}$) currently ~~represents~~ $\sim 25\%$ of the total stratospheric bromine loading. Owing to ~~its~~ much shorter lifetime compared to anthropogenic long-lived bromine (e.g. halons) and chlorine (e.g. chlorofluorocarbons), the impact of VSL$^{Br}$ on ozone peaks in the lowermost stratosphere, which is a key climatic and radiative atmospheric region. Here we present a modelling study of the evolution of stratospheric ozone and its chemical loss within the tropics and ~~at~~ mid-latitudes during the 21st century. Two different experiments are explored: considering and neglecting the additional stratospheric injection of 5 ppt biogenic VSL$^{Br}$ naturally released from the ocean. Our analysis shows that the inclusion of VSL$^{Br}$ results in a realistic stratospheric bromine loading and improves agreement between the model and satellite observations of the total ozone column (TOC) for the 1980–2015 period at mid-latitudes. We show that the overall ozone response to VSL$^{Br}$ at mid-latitudes follows the stratospheric evolution of long-lived inorganic chlorine and bromine throughout the 21st century. Additional ozone loss due to VSL$^{Br}$ is maximized during the present-day period (1990–2010), with TOC differences of $-8\,\mathrm{DU}$ ($-3\%$) and $-5.5\,\mathrm{DU}$ ($-2\%$) for the Southern Hemisphere (~~SH~~) and Northern Hemisphere (~~NH~~) mid-latitudes (~~MLs[CE1]~~), respectively. Moreover, the projected TOC differences at the end of the 21st century are

$\sim 50\%$ lower than the values found for the present-day period.

~~We find that seasonal VSL$^{Br}$ impacts on lowermost mid-latitude stratospheric ozone are influenced by the seasonality of heterogeneous inorganic chlorine reactivation processes in ice crystals.~~ Indeed, due to the more efficient reactivation of chlorine reservoirs (mainly ClONO$_2$ and HCl) within the colder ~~lowermost stratosphere at SH MLs,~~ the seasonal VSL$^{Br}$ impact shows a small but persistent hemispheric asymmetry through the whole modelled period. Our results indicate that, although the overall VSL$^{Br}$-driven ozone destruction is greatest during spring, the halogen-mediated (Halog$_{x\text{-Loss}}$) ozone loss cycle in the ~~lowermost mid-latitude stratosphere~~ during winter is comparatively more efficient than the HO$_x$ cycle with respect to other seasons. Indeed, when VSL$^{Br}$ ~~is~~ considered, Halog$_{x\text{-Loss}}$ dominates wintertime lowermost stratospheric ozone loss at ~~SH MLs~~ between 1985 and 2020, with a contribution of inter-halogen ClO$_x$–BrO$_x$ cycles to Halog$_{x\text{-Loss}}$ of $\sim 50\%$.

Within the tropics, a small ($< -2.5\,\mathrm{DU}$) and relatively constant ($\sim -1\%$) ozone depletion mediated by VSL$^{Br}$ is closely related to its fixed emissions throughout the modelled period. By including the VSL$^{Br}$ sources, the seasonal Halog$_{x\text{-Loss}}$ contribution to lowermost stratospheric ozone loss is practically dominated by the BrO$_x$ cycle, reflecting the low sensitivity of ~~VSL$^{Br}$~~ to background halogen abundances,

~~which allows it to drive tropical stratospheric ozone depletion.~~ We conclude that, considering the coupling between biogenic bromine sources and seasonal changes of the chlorine, heterogeneous reactivation is a key feature for future projections of ~~lowermost mid latitude stratospheric~~ ozone during the 21st century.

---

## 1 Introduction

The role of bromine in stratospheric ozone depletion has been discussed in several studies (Wofsy et al., 1975; Prather and Watson, 1990; Daniel et al., 1999; Dvortsov et al., 1999; Solomon, 1999 [TS2]). Although bromine is much less abundant than chlorine in the atmosphere, it is known to deplete stratospheric ozone 45 to 69 times more efficiently on a per atom basis (Daniel et al., 1999; Sinnhuber et al., 2009). Moreover, the ozone depletion efficiency of active bromine (Br and BrO) is strongly related to the available amount of activated chlorine (mainly Cl and ClO radicals) in the atmosphere (McElroy et al., 1986; Solomon et al., 1999; and references therein [CE2]) as well as to enhanced sulfate aerosol loading via heterogeneous reactions (Salawitch et al., 2005). Consequently, even low concentrations of bromine have a relatively large impact on stratospheric ozone. In addition to anthropogenic long-lived chlorine ($LL^{Cl}$) and bromine ($LL^{Br}$) substances such as chlorofluorocarbons (CFCs), chlorocarbons, methyl bromide, and bromofluorocarbons (halon fire suppressants), other substances with photochemical lifetimes shorter than 6 months, often referred to as very ~~short-lived~~ (VSL) substances, have the potential to transport a significant amount of reactive halogens into the stratosphere. Owing to their short lifetimes, the impact of VSL substances on stratospheric ozone peaks in the lowermost stratosphere (Salawitch et al., 2005; Feng et al., 2007; Sinnhuber et al., 2009; Sinnhuber and Meul, 2015 [TS3]; Yang et al., 2014; Hossaini et al., 2015a; Falk et al., 2017), which is an important atmospheric region because surface temperature and climate are most sensitive to ozone perturbations (Riese et al., 2012; Hossaini et al., 2015b; Iglesias-Suarez et al., 2018). In fact, Hossaini et al. (2015b) found that ~~very short lived bromine~~ ($VSL^{Br}$) exerts an ozone radiative effect (normalized by halogen content) that is 3.6 times larger than that arising from long-lived substances.

$VSL^{Br}$ ~~is~~ produced in the ocean via the metabolism of marine organisms such as phytoplankton and macroalgae (e.g. Carpenter et al., 2007; Quack et al., 2007; Leedham Elvidge et al., 2015), even in sea ice (e.g. Abrahamsson et al., 2018). Due to ~~its~~ high volatility, ~~it is~~ transferred into the marine boundary layer through air–sea exchange (Carpenter and Liss, 2000; Quack and Wallace, 2003). The most abundant $VSL^{Br}$ ~~compounds~~ released to the atmosphere are bromoform ($CHBr_3$) and dibromomethane (methylene dibromide, $CH_2Br_2$), followed

by a minor (but non-negligible) contribution of interhalogen species (bromochloromethane, $CH_2BrCl$; dibromochloromethane, $CHBr_2Cl$; and bromodichloromethane, $CHBrCl_2$). Altogether, $VSL^{Br}$ ~~contributes~~ $\sim 5$ (3–7) ppt to stratospheric bromine, which accounted for about 25 % of total stratospheric bromine in 2016 (WMO, 2018). Moreover, this additional input of bromine is required to reconcile current stratospheric bromine trends (Salawitch et al., 2010; WMO, 2018).

Production of anthropogenic $LL^{Cl}$ and $LL^{Br}$ substances has been restricted by the Montreal Protocol and its subsequent amendments and adjustments (Solomon, 1999 [TS4]; WMO, 2018). After application, the stratospheric $LL^{Cl}$ and $LL^{Br}$ load peaked at the end of the 20th century and has been decreasing at a rate that depends on their respective atmospheric lifetimes. This implies, for example, that hydrogen chloride (HCl) shows a long-term decrease at a rate of about 0.5 % $yr^{-1}$ in the middle stratosphere between 60° N and 60° S, while total stratospheric bromine decreased at a rate of about 0.75 % $yr^{-1}$ from 2004 to 2014 (WMO, 2018). Accordingly, stratospheric ozone is expected to recover from the effects of anthropogenic halogen-induced loss on a similar timescale, which is already detectable in the Antarctic and the upper stratosphere (e.g. Solomon et al., 2016a; Chipperfield et al., 2017; Dhomse et al., 2018; Strahan and Douglass, 2008; and references therein). Hossaini et al. (2015a) quantified the stratospheric injection of organic and inorganic chlorine from anthropogenic $VSL^{Cl}$ sources such as chloroform ($CHCl_3$), dichloromethane ($CH_2Cl_2$), tetrachloroethene ($C_2Cl_4$), trichloroethene ($C_2HCl_3$), and 1,2-dichloroethane ($CH_2ClCH_2Cl$), which are not controlled by the Montreal Protocol. From their results, the total stratospheric chlorine load from $VSL^{Cl}$ inferred for 2013 is 123 ppt, with a stratospheric injection dominated by source gases ($\sim 83$ %). Moreover, the stratospheric ~~$VSL^{Cl}$~~ (organic and inorganic) injection increased by $\sim 52$ % between 2005 and 2013, mainly due to recent and ongoing growth in anthropogenic $CH_2Cl_2$ emissions. In fact, Hossaini et al. (2017) showed that the impact of $CH_2Cl_2$ on stratospheric ozone has increased markedly in recent years, and if these increases continue into the future, the recovery of the ~~earth's~~ ozone layer could be delayed even further, offsetting some of the gains achieved by the Montreal Protocol.

The total stratospheric chlorine and bromine budgets derived for 2016 were 3.29 ppb Cl and 19.60 ppt Br, respectively (WMO, 2018), and are expected to return to their 1980 values, an arbitrary reference date before the discovery of the Antarctic ozone hole, around the middle of this century (Dhomse et al., 2018). Although we still lack scientific consensus with respect to the future evolution of $VSL^{Br}$ oceanic source strength and stratospheric injection (WMO, 2014; Lennartz et al., 2015; Ziska et al., 2017), the continuous decrease ~~in~~ $LL^{Br}$ leads to an increase in the relative $VSL^{Br}$ contribution to total stratospheric bromine into the fu-

Please note the remarks at the end of the manuscript.

ture. Thus, understanding the role of natural VSL[Br] sources is a key issue for chemistry–climate projections.

The impact of additional bromine is closely related to the heterogeneous chemistry of chlorine (Solomon, 1999 [TS5]; Salawitch et al., 2005; Müller, 2012). The influence of temperature and water vapour in the heterogeneous conversion processes of inorganic-chlorine reservoirs (mostly HCl and ClONO$_2$) into active radicals on sulfate aerosols has been reviewed elsewhere (e.g. Anderson et al., 2012, 2017; Drdla and Müller, 2012; Anderson and Clapp, 2018). In particular, Drdla and Müller (2012) highlighted that an increase in water vapour above background values would allow chlorine activation at higher temperatures than those observed in polar regions, which led to the hypothesis that chlorine reactivation and subsequent ozone loss could occur in the lower stratosphere at mid-latitudes during summer (Robrecht et al., 2019). Indeed, the spatial and seasonal distributions of chlorine monoxide and chlorine nitrate in the monsoon regions provide indicators of heterogeneous chlorine processing in the tropical and subtropical lowermost stratosphere of the Northern Hemisphere (Solomon et al., 2016b). Moreover, heterogeneous chorine reactivation has also been observed in ice particles in cirrus clouds located near the tropopause (Borrman et al., 1996, 1997; Solomon et al., 1997; Bregman et al., 2002; Thornton et al., 2003). The occurrence of cirrus clouds was reported in the lowermost stratosphere of northern high-mid-latitudes (40–75° N; Spang et al., 2015), with summertime clouds located up to ∼ 3 km (or 40–50 K of potential temperature) above the tropopause (Dessler, 2009). Furthermore, von Hobe et al. (2011) suggested that, at low temperatures, cirrus ice particles may promote a significant activation of heterogeneous chlorine in the tropical upper troposphere and lower stratosphere. Finally, satellite observations indicate that chlorine activation also occurs in the Antarctic and Arctic sub-vortex regions, where processed air dispersing from the decaying vortex in spring induces rapid changes in extra-vortex trace gas abundances (Santee et al., 2011).

Previous chemistry–climate modelling studies considering VSL[Br] chemistry have mainly focused on improving the model vs. observed mid-latitude ozone trends at the end of the 20th century (Feng et al., 2007; Sinnhuber et al., 2009). More recently, Sinnhuber and Meul (2015) have shown much greater ozone depletion in the lowermost stratosphere during the 1979–1995 period and a greater ozone increase during the 1996–2005 period, which is in better agreement with observations due to the inclusion of VSL[Br] sources. Moreover, Falk et al. (2017) have reported that the projected additional VSL[Br] impact on ozone at the end of the 21st century is reduced compared to present day. These results are in agreement with those of Yang et al. (2014), who performed time slice simulations to address the sensitivity of stratospheric ozone to a speculative doubling of VSL[Br] sources under different LL[Cl] scenarios. However, neither of the previous studies had distinguished the contribution of ozone depletion –

including its seasonal variability – that arises from VSL[Br] with respect to LL[Cl] and LL[Br] throughout the 21st century.

In this work, using the CAM-chem model, we present a coupled chemistry–climate modelling study from 1960 to 2100 with and without the contribution from VSL[Br] sources. We focus on natural VSL[Br]-driven impacts on the temporal evolution of stratospheric ozone and the total ozone column (TOC) in the tropics and at mid-latitudes, distinguishing both the long-term seasonal change and the resulting hemispheric asymmetries. Additionally, we present a timeline assessment of the individual contribution from anthropogenic (LL[Cl] and LL[Br]) and biogenic (VSL[Br]) sources to stratospheric halogen loading throughout the 21st century while also recognizing the VSL[Br] contribution to overall halogen-catalysed ozone loss in the lowermost stratosphere. The layout of the paper is as follows: Sect. 2 describes the main characteristics and configuration of the CAM-chem model as well as the observation data sets that are used for the evaluation of the CAM-chem performance. Section 3 presents a quantitative comparison between the model and observations of stratospheric chlorine and bromine loading (Sect. 3.1) and the TOC (Sect. 3.2) as well as discusses long-term seasonal impacts mediated by VSL[Br] on the TOC (Sect. 3.2) and the lowermost stratospheric ozone budget (Sect. 3.3). Changes in the seasonal evolution of halogen-driven ozone loss rates due to VSL[Br] are shown in Sect. 3.4. The final concluding remarks are summarized in Sect. 4.

## 2 Methods: model description

The Community Earth System Model (CESM1) with the Community Atmospheric Model including interactive chemistry (CAM4-chem version; Lamarque et al., 2012; Tilmes et al., 2016) has been used to explore VSL[Br]-driven stratospheric ozone loss in three latitude bands: Southern Hemisphere mid-latitudes (SH MLs; 60–35° S), tropics (25° N–25° S), and Northern Hemisphere mid-latitudes (NH MLs; 35–60° N). The model set-up follows the Chemistry–Climate Model Initiative (CCMI) REFC2 configuration described in detail by Tilmes et al. (2016), with the exception that in this work we consider a full halogen chemistry mechanism and oceanic emission – seasonally dependent and geographically distributed – of six bromocarbons (VSL[Br] = CHBr$_3$, CH$_2$Br$_2$, CH$_2$BrCl, CHBrCl$_2$, CHBr$_2$Cl, and CH$_2$IBr) from the Ordóñez et al. (2012) emission inventory. This emission inventory is based on a monthly varying satellite chl a climatology, which allows the introduction of a complete seasonal cycle to the emission strength and spatial distribution of oceanic bromocarbons. A full description of the VSL[Br] mechanism implemented in CAM-chem – including both biogenic and anthropogenic sources, heterogeneous recycling reactions, dry and wet deposition, convective uplift, and large-scale transport – has been given elsewhere (Ordóñez et al., 2012; Fernandez et al., 2014). Monthly and sea-

sonally varying lower boundary conditions were considered for long-lived chlorine ($LL^{Cl} = CH_3Cl$, $CH_3CCl_3$, $CCl_4$, CFC-11, CFC-12, CFC-113, HCFC-22, CFC-114, CFC-115, HCFC-141b, and HCFC-142b) and long-lived bromine ($LL^{Br} = CH_3Br$, H-1301, H-1211, H-1202, and H-2402) following the A1 halogenated ozone-depleting substance emissions scenario ~~from the WMO (2011) Ozone Assessment Report~~. The surface concentrations of $CO_2$, $CH_4$, $H_2$, and $N_2O$ were specified following the moderate Representation Concentration Pathway 6.0 (RCP6.0) scenario (Meinshausen et al., 2011; Eyring et al., 2013).

The ~~CAM-chem~~ configuration used in this work extends from the surface to approximately 40 km (3.5 hPa in the upper stratosphere), with 26 vertical levels, and includes a horizontal resolution of 1.9° latitude by 2.5° longitude. The number of stratospheric levels changes depending on the latitudinal region: within the tropics, there are eight levels above the tropopause ($\sim 100$ hPa), with a mean thickness of 1.25 km (15.5 hPa) for the lower stratospheric levels and 5.2 km (3.8 hPa) between the two highest levels, while for the mid-latitudes, the tropopause is located at approximately $\sim 200$ hPa, and the stratosphere contains up to 12 levels. To ensure a consistent dynamical description of the stratosphere within the relatively low model top of ~~CAM-chem~~ (i.e. gravity wave dragging and the Brewer–Dobson circulation), the integrated momentum that would have been deposited above the model top is specified by an upper boundary condition (Lamarque et al., 2008). A similar procedure is applied to the altitude-dependent photolysis rate computations, which include an upper boundary condition that considers the ozone column fraction prevailing above the model top. The CAM4-~~chem~~ version used in this work includes a non-orographic gravity wave scheme based on the inertia–gravity wave parameterization and an observation-based implementation of stratospheric aerosol and surface area density. The quasi-biennial oscillation is imposed in the model by relaxing equatorial zonal winds between 90 and 3 hPa to the observed interannual variability (see Tilmes et al., 2016, for more details).

Equivalent ~~CAM-chem~~ configurations, such as the one implemented in this work, have been used for the CCMVal-2 (Chemistry–Climate Model Validation 2) and CMIP5 (Coupled Model Intercomparison Project, Phase 5) activities in order to represent trends in the ozone evolution and to estimate the return dates, which lie in the middle of the multi-model range (Eyring et al., 2010, 2013). Moreover, our model configuration uses a fully coupled ~~earth system model~~ approach, i.e. the ocean and sea ice are explicitly computed (Neale et al., 2013). This implies that instead of isolating the chemical impact mediated by $VSL^{Br}$ ~~sources~~ on the climatological ozone budget (as would be the case using a specified dynamic approach), the chemical interaction between ~~$VSL^{Br}$~~ and ozone in the lowermost stratosphere is affected by dynamic feedbacks (i.e. temperature, radiation, etc.).

Even though the interactive ocean coupling would have allowed us to compute the evolution of $VSL^{Br}$ ocean source emissions throughout the modelled period, we still lack a considerable understanding of the seasonal processes that dominate $VSL^{Br}$ ocean emissions in both the present and the future, which is combined with a limited observation data set of VSL bromocarbons (WMO, 2014, 2018). Therefore, in this work, we forced the model with annually cycled $VSL^{Br}$ fluxes, replicating the Ordoñez et al. emission inventory for all years between 1950 and 2100. This procedure aims to reduce the uncertainties associated with the unknown evolution of both biogeochemical production and oceanic emissions of $VSL^{Br}$ into the future (Lennartz et al., 2015; Ziska et al., 2017).

Two ensembles of independent modelled experiments were performed, each with three individual simulations from 1960 to 2100, considering approximately 10 years of spin-up to allow stabilization of the stratospheric circulation. The individual simulations 001, 002, and 003 started in 1950, 1951, and 1952, respectively, with initial conditions taken from the equivalent years of the CESM1 WACCM (Whole Atmosphere Community Climate Model) 20th-century ensemble for CMIP5 (Marsh et al., 2013). The baseline simulation set-up of the first experiment considered only the halogen $LL^{Cl}$ and $LL^{Br}$ contribution from anthropogenic chlorofluorocarbons, hydrochlorofluorocarbons, halons, methyl chloride, and methyl bromide, while the simulation set-up of the second experiment included the background biogenic contribution from $VSL^{Br}$ oceanic sources in addition to the anthropogenic sources above. The mean ensemble of each experiment (i.e. with and without the $VSL^{Br}$ sources) was calculated as the average of the three individual simulations. Differences between these two mean ensembles allow quantification of the overall impact of $VSL^{Br}$ ~~sources~~ on stratospheric ozone, ~~and they allow us~~ to determine their relative contributions to the halogen-mediated catalytic ozone-depleting families. In addition, from this analysis it is also possible to distinguish the contributions of both $LL^{Br}$ and $VSL^{Br}$ to the inorganic fraction of stratospheric bromine ($Br_y^{LL+VSL} = Br_y^{LL} + Br_y^{VSL} = Br + BrO + HBr + BrONO_2 + BrCl + HOBr + 2Br_2 + BrNO_2$), whereas the inorganic fraction of chlorine ($Cl_y^{LL} = ClO + Cl + HOCl + 2Cl_2O_2 + 2Cl_2 + OClO + HCl + ClONO_2 + BrCl + ClONO_2$) arises only from the degradation of $LL^{Cl}$, which is identical for both experiments.

For the case of vertical distributions and latitudinal variations, the zonal mean of each mean ensemble was computed from the monthly output before processing the data, while a locally weighted scatter plot smoothing (LOWESS) filter with a 0.2 fraction was applied to all long-term time series to smooth the data. Most of the figures and values in the Results section include geographically averaged quantities for the present day (defined here as the 1990–2010 mean, nominally year 2000) and the end of the 21st century (defined here as the 2080–2100 mean, nominally year 2100) over the latitu-

dinal bands defined above. To highlight seasonal changes, we computed the average for the months of December–January–February (DJF), March–April–May (MAM), June–July–August (JJA), and September–October–November (SON). Ozone loss trends due to VSL$^{Br}$ between 2000 and 2100 were calculated on the basis of the least-squares linear trends, while the trend errors were estimated as twice the statistical deviation stemming from the least-squares fit.

To provide a basic evaluation of the model performance, the modelled results of the TOC and the inorganic-halogen abundances were compared with the following selected observation data sets:

i. the Microwave Limb Sounder (MLS) observations of the annual mean volume mixing ratio of HCl + ClO (Waters et al., 2006; Livesey et al., 2018), which provides a lower limit of the Cl$_y$ abundance in the stratosphere from 2005 to 2015;

ii. the Br$_y$ trend reported in the latest WMO Ozone Assessment Reports (WMO, 2014, 2018), which shows changes in total Br$_y$ between 1991 and 2012 derived from stratospheric BrO observations by balloon-borne (Dorf et al., 2006) and ground-based UV–visible (at Harestua, 60° N, and Lauder, 45° S) measurements (Hendrick et al., 2007, 2008);

iii. the Solar Backscatter Ultraviolet (SBUV) merged total ozone column data set (MOD; Frith et al., 2014, 2017) from 1980 to 2015.

An important point to highlight about the SBUV MOD (and its uncertainties) is that it was constructed from ozone profiles measured by individual SBUV instruments and retrieved using the Version 8.6 algorithm (Bhartia et al., 2013; Frith et al., 2014, 2017). To maintain consistency over the entire time series, the individual instrument records were analysed with respect to each other, and absolute calibration adjustments were applied as needed based on comparison of radiance measurements during periods of instrument overlap (DeLand et al., 2012; Weber et al., 2018 [TS4]). During the overlap periods, the records of the involved SBUV instruments were combined using a simple average. Frith et al. (2014) have estimated the potential errors in the SBUV MOD through Monte Carlo model simulations, taking into account the range of offsets and drifts observed during the overlap periods between individual SBUV instruments and between SBUV and Dobson ground-based measurements. The authors also assessed the extent to which these potential errors might affect the long-term variability of ozone in a multiple-regression model. See Frith et al. (2014) for a detailed description of the data used and their associated uncertainties in the SBUV MOD.

## 3   Results and discussion

### 3.1   Contribution of LL$^{Cl}$, LL$^{Br}$, and VSL$^{Br}$ to stratospheric inorganic-halogen loading

The evolution (1960–2100) of the modelled annual mean abundances of inorganic chlorine (Cl$_y$) and bromine (Br$_y$) as well as the VSL$^{Br}$ sources in the global upper stratosphere (3.5 hPa) and lower stratosphere (50 hPa) at mid-latitudes and in the tropics are shown in Fig. 1. In addition, Fig. 1 includes global Br$_y$ trends reported in recent WMO reports (WMO, 2014, 2018) and the MLS observations (HCl + ClO). The values of bias, normalized mean bias (NMB), and normalized mean error (NME) used to evaluate the agreement between the modelled Cl$_y$ and observations for each of the regions under study are shown in Table S1 in the Supplement. Clearly, the temporal evolution of Cl$_y^{LL}$ and Br$_y^{LL}$ shows a pronounced peak at the end of the 20th century and beginning of the 21st century, respectively, after which the abundance of both halogens declines. Within the A1 halogen emission scenario considered in this work, the Cl$_y^{LL}$ abundance in our model returns to its past 1980 levels just before ∼ 2060 in the global upper stratosphere, whereas the prevailing Br$_y^{LL}$ abundance for the year 1980 is not recovered even by the end of the 21st century (see Fig. 1a). In comparison, the evolution of Br$_y^{VSL}$ abundances remains constant over time, resulting in an additional and time-independent fixed contribution of ∼ 5 ppt total bromine injection, which leads to maximum global upper-stratosphere Br$_y^{LL+VSL}$ abundances of up to ∼ 20.5 ppt at the beginning of the 21st century. This is in agreement with previous studies performed only for present-day conditions (Fernandez et al., 2014). Furthermore, note that the modelled global Br$_y^{LL+VSL}$ abundances are in good agreement with the observations within reported errors, highlighting the importance of considering the additional contribution from VSL$^{Br}$ to determine the overall evolution of stratospheric ozone during the 21st century.

In the lower stratosphere, at 50 hPa, Br$_y^{LL}$ abundances return to 1980 levels before ∼ 2080 and ∼ 2050 for the mid-latitudes and tropics, respectively, while the Cl$_y^{LL}$ return date remains unaltered (see Fig. 1b–d). Furthermore, the contribution from VSL$^{Br}$ to mid-latitude total inorganic bromine also reaches ∼ 5 ppt throughout the modelled period, while in the tropics there prevails a small carbon-bonded organic fraction that has not been converted to the reactive inorganic form. Falk et al. (2017) showed an increase in VSL$^{Br}$ of about 0.5 ppt in the tropical lowermost stratosphere by the end of the century compared to present day under the RCP6.0 scenario and assuming constant VSL$^{Br}$ fluxes, following scenario five of Warwick et al. (2006). This increase in the stratospheric VSL$^{Br}$ attributed to enhanced vertical transport in the tropics is counteracted by a decrease in the Br$_y^{VSL}$ injected into the stratosphere so that the total stratospheric VSL$^{Br}$ remains unchanged between the two periods. In this work, the stratospheric injection of VSL$^{Br}$ and Br$_y^{VSL}$

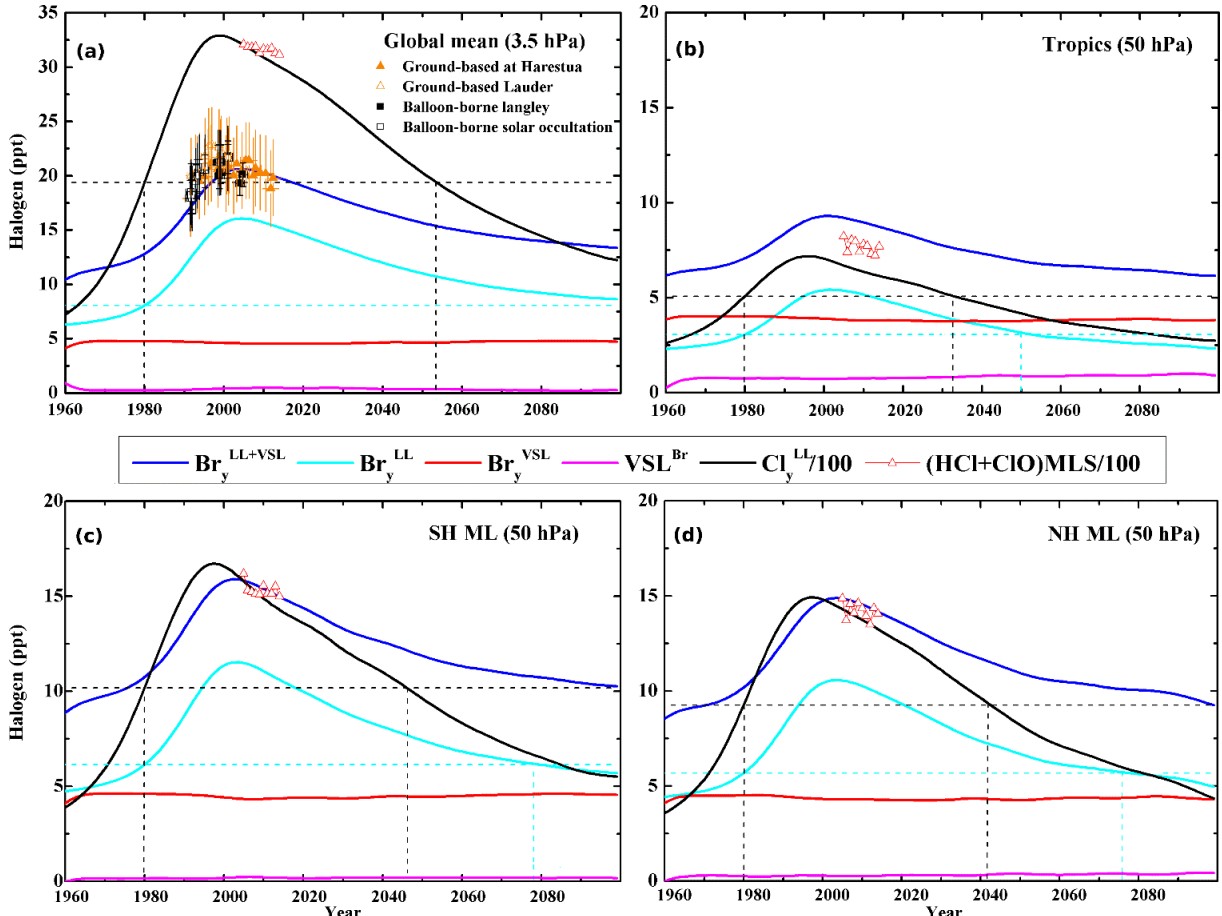

**Figure 1.** Temporal evolution (1960–2100) of the modelled annual mean abundances of $VSL^{Br}$ and inorganic halogen ($Cl_y^{LL}$ and $Br_y^{LL+VSL}$): **(a)** global upper stratosphere (3.5 hPa) and lower stratosphere (50 hPa) **(b)** in the tropics and at **(c)** SH MLs and **(d)** NH MLs. The $Br_y^{LL+VSL}$ abundance was split into long-lived ($Br_y^{LL}$) and very short-lived ($Br_y^{VSL}$) contributions. The black squares (filled and open) and orange triangles (filled and open) of **(a)** show the total $Br_y$ evolution (1991–2012) reported in the latest WMO Ozone Assessment Reports (2014, 2018). The red triangles show the mean annual mixing ratios (HCl + ClO) from the Microwave Limb Sounder (MLS) data (Waters et al., 2006; Livesey et al., 2018) from 2005 to 2015. The dashed horizontal lines indicate the modelled $Cl_y^{LL}$ and $Br_y^{LL}$ abundances for 1980. Note that both $Cl_y^{LL}$ values and (HCl + ClO) MLS data were divided by 100.

between 2000 and 2100 remains unchanged. Based on the limited observational data, it is uncertain whether any trend in the stratospheric injection of organic and inorganic $VSL^{Br}$ would be expected, and thus no conclusive statement can be made. Nevertheless, our present-day stratospheric injection of $VSL^{Br}$ (∼ 2 ppt) and $Br_y^{VSL}$ (∼ 3 ppt) is in perfect accordance with that reported in the latest WMO Ozone Assessment Report (2018). Further studies are needed to evaluate the uncertain evolution of the stratospheric injection of organic and inorganic species from $VSL^{Br}$ sources throughout the 21st century.

The modelled inorganic-chlorine abundance shows a good agreement with the (HCl + ClO) MLS observations for the 2005–2015 period mainly in the upper stratosphere, where most chlorine has already been photochemically converted to $Cl_y^{LL}$. For example, the normal mean error of the compar-

ison between modelled $Cl_y^{LL}$ abundances and MLS observations is about 3.5 % and 19 % at 50 hPa for the mid-latitudes and tropics, respectively, while the NME is about 2 % for the global mean at 3.5 hPa (see Table S1). This relatively good agreement between the model and MLS observation occurs even without consideration of the recent contribution of anthropogenic $VSL^{Cl}$ sources (i.e. Hossaini et al., 2015b, 2017). Moreover, the inter-annual variation in MLS observations at 50 hPa occurs mainly due to heterogeneous chlorine reactivation, which can be influenced by the polar vortex dynamics at mid-latitudes (Dhomse et al., 2018). Note that although we processed output at 120 hPa in the lowermost mid-latitude stratosphere in Sect. 3.4, we did not compare $Cl_y^{LL}$ at this altitude level. This is because below 50 hPa, the fractional conversion of organic chlorine to $Cl_y^{LL}$ is very small, and thus the inferred HCl + ClO MLS data have a very large

uncertainty that prevents a reliable ~~model observation inter-comparison~~ (Santee et al., 2011).

The stratospheric $Br_y^{VSL}$ injection remains constant during the whole modelled period, representing $\sim 25\%$ of $Br_y^{LL+VSL}$ in the global upper stratosphere for the year 2000 and increasing up to $\sim 40\%$ by the end of the 21st century. These results are in agreement with those reported by Fernandez et al. (2017). Due to the much shorter lifetimes of $VSL^{Br}$ with respect to $LL^{Br}$, the contribution percentage is larger in the lower stratosphere. For example, the $Br_y^{VSL}$ relative contribution to $Br_y^{LL+VSL}$ for the year 2000 at 50 hPa represents $\sim 30\%$ and $\sim 45\%$ for the mid-latitudes and tropics, respectively. Furthermore, at mid-latitudes, $Br_y^{VSL}$ represents up to 45 % of $Br_y^{LL+VSL}$ both at the end of the 21st century and in the years prior to 1980, while in the tropics $Br_y^{VSL}$ represents up to 65 % for these time periods. Consequently, these changes in the relative contribution of $VSL^{Br}$ to total inorganic bromine with respect to time, region, and altitude in the presence of different inorganic-chlorine abundances lead to relatively different ozone loss within those regions and altitude regimes, as described in detail below.

Figure 2 shows the annual zonal mean distribution of ~~inorganic chlorine~~ and CE7 bromine abundances as well as the corresponding $ClO_x/Cl_y$ and $BrO_x/Br_y$ percentage ratios (black contour lines) for the present day. Note that the $ClO_x/Cl_y$ and $BrO_x/Br_y$ ratios reflect the changes in the contribution of the active-chlorine ($ClO_x = ClO + Cl + HOCl + 2Cl_2O_2 + OClO + 2Cl_2$) and CE8 ~~bromine~~ ($BrO_x = BrO + Br$) fractions relative to their total inorganic abundances. Equivalent results are presented for the end of the 21st century in the Supplement (see Fig. S1). In the lowermost stratosphere, the available inorganic fraction of both chlorine and bromine rapidly increases with altitude as halogen atoms are released by the photolysis and oxidation from all $LL^{Cl}$, $LL^{Br}$, and $VSL^{Br}$ species. In contrast, within the upper stratosphere no major changes in inorganic-halogen abundances are observed since the conversion from their organic sources is nearly complete. Although our model shows a symmetric hemispheric distribution in stratospheric $Cl_y$ and $Br_y$ abundances, there is a marked difference in the $ClO_x/Cl_y$ ratio. In fact, the experiments capture a local increase in the $ClO_x/Cl_y$ ratio at 15° N and $\sim 100$ hPa as a result of a large increase in ~~$Cl_y$ species' heterogeneous reactivation due to~~ enhanced vertical transport during the Indian summer monsoon ~~season~~ (Solomon et al., 2016b). Along the same lines, the higher $ClO_x/Cl_y$ values modelled in the lowermost stratosphere at ~~SH MLs~~ compared to ~~NH MLs~~ are mainly due to the enhancement in the heterogeneous chlorine reactivation processes ~~in~~ ice crystals (i.e. HCl and $ClONO_2$ reactivation; Santee et al., 2011; Solomon, 1999 TS7; and references therein), influenced mainly by the proximity of the Antarctic polar vortex edge and the lower temperatures prevailing at high and mid-latitudes of the Southern Hemisphere (see Fig. S2). Additionally, the favourable conditions for

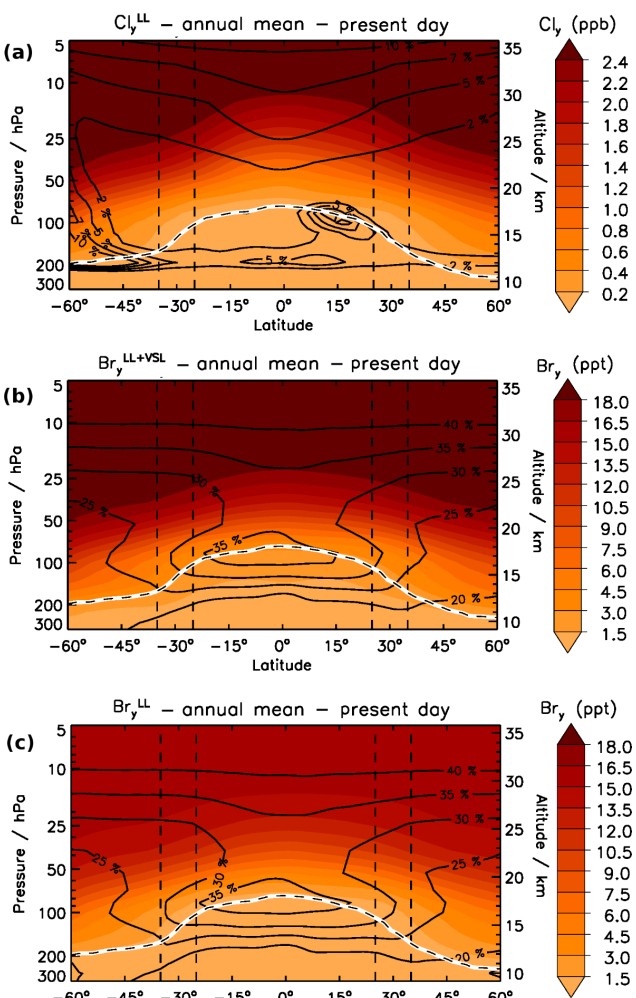

**Figure 2.** Annual zonal mean distribution of **(a)** $Cl_y^{LL}$, **(b)** $Br_y^{LL+VSL}$, and **(c)** $Br_y^{LL}$ during the present-day period. The colour scale represents volume mixing ratios (ppb or ppt), while black contour lines show the contribution percentage of $ClO_x$ to $Cl_y$ and $BrO_x$ to $Br_y$. The lower solid white line indicates the location of the tropopause (chemical definition of 150 ppb ozone level from the experiment without $VSL^{Br}$ sources).

isentropic exchange of young subtropical air with extremely old winter polar vortex air masses at ~~SH MLs~~ (Spang et al., 2015; Rolf et al., 2015) could drive additional heterogeneous chlorine reactivation and thus enhance hemispheric asymmetry. Moreover, the increase in the $ClO_x$ abundance and $ClO_x/Cl_y$ ratio is highest during winter and spring for both ~~SH MLs and NH MLs~~, which is consistent with the seasonal changes of the lowermost stratospheric temperatures that affect chlorine reactivation ~~in~~ ice crystals (see Fig. S3). In contrast, a symmetric hemispheric distribution of the $BrO_x/Br_y$ ratio is modelled in the ~~lowermost mid-latitude stratosphere~~ and is associated with heterogeneous bromine reactivation ~~in~~ sulfate aerosols instead of ice crystals (i.e. HBr and $BrONO_2$

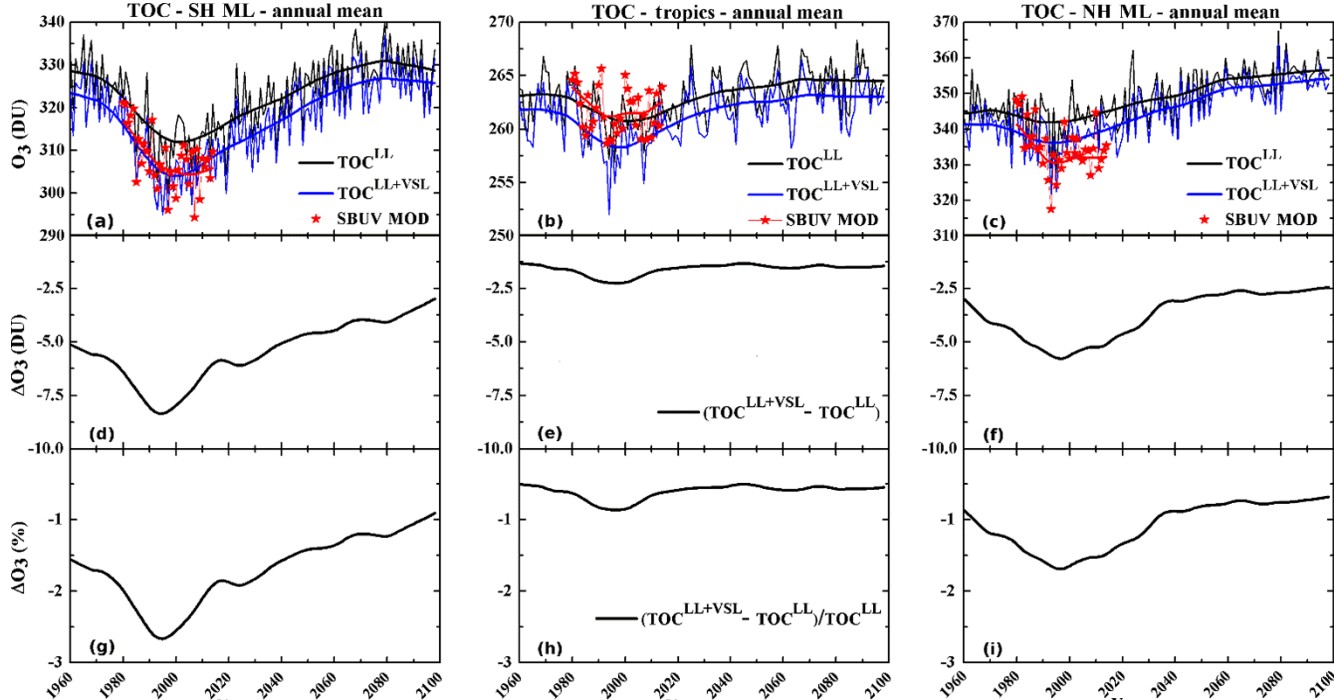

**Figure 3.** Temporal evolution of the annual mean total ozone column ($TOC^{LL+VSL}$ and $TOC^{LL}$) **(a)** at SH MLs, **(b)** in the tropics, and **(c)** at NH MLs as well as the corresponding absolute (DU; **d–f**) and relative (%; **g–i**) TOC difference ($\Delta TOC$). TOC values of the mean ensemble (thin lines) and the smoothed time series via LOWESS filtering (0.2 fractions; thick lines) are shown in blue for $TOC^{LL+VSL}$ and black for $TOC^{LL}$. The red lines and symbols show the observations from the Solar Backscatter Ultraviolet (SBUV) merged total ozone column data set (Frith et al., 2014, 2017) within the same spatial and temporal mask as the model output.

reactivity; Solomon, 1999, TS8 and references therein), which show neither any hemispheric asymmetry nor seasonal changes in the reactivation processes (see Fig. S4). Thus, even when the inclusion of $VSL^{Br}$ sources increases the total stratospheric $Br_y$ abundance and therefore its active fraction ($BrO_x$), the $BrO_x/Br_y$ ratio remains unchanged in the stratosphere during the whole modelled period (see Figs. 2b, c and S1b, c), highlighting that this ratio is nearly independent of the total inorganic bromine abundance.

## 3.2   Impact of $VSL^{Br}$ on the seasonal evolution of the total ozone column (TOC)

The temporal evolution (1960–2100) of the modelled annual mean TOC at mid-latitudes and in the tropics, along with SBUV MOD TOC observations, is illustrated in Fig. 3. Equivalent results for the temporal evolution of the individual simulations of each experiment are shown in Fig. S5. Table 1 shows the annual and seasonal mean values of the absolute and relative TOC differences ($\Delta TOC$) between the experiments for the present day and the end of the 21st century as well as the $\Delta TOC$ trends (% per decade) over the century. In addition, Table S2 shows the values of bias, normalized mean bias (NMB), and normalized mean error (NME) to evaluate the agreement between the modelled TOC and observations

for each of the regions under study. Based on the comparison of the TOC results over the different latitudinal bands, the following features can be described:

i.  The constant emission of biogenic $VSL^{Br}$ sources introduces a continuous reduction in the TOC that exceeds the mean ensemble variability between the experiments (see Fig. S5). Moreover, the inclusion of $VSL^{Br}$ sources improves the overall agreement between the model and observations, with a reduction of 1 % and 0.9 % in NME for SH MLs and NH MLs, respectively (see Table S2). However, the inclusion of $VSL^{Br}$ sources leads to a slight increase of 0.2 % in NME for the tropics. Note that the statistical analysis presented in Table S2 does not take into account the uncertainties that may arise from the merging of the individual SBUV instruments in the SBUV MOD (see Sect. 2).

ii. At mid-latitudes, the additional ozone loss due to $VSL^{Br}$ peaks during the present-day period, coinciding with the temporal location of the minimum TOC for both experiments and reaching a $\Delta TOC^{2000}$ of approximately $-8$ DU ($\sim -2.5$ %) and $-5.5$ DU ($\sim -1.6$ %) for SH MLs and NH MLs, respectively (see Table 1). Moreover, $VSL^{Br}$ impacts on the TOC by the end of the 21st century are $\sim 50$ % lower than the values found

**Table 1.** Annual and seasonal TOC changes ($\Delta$TOC) mediated by VSL$^{Br}$ ~~at mid-latitudes and in the tropics during the present-day~~ period and the end of the 21st century as well as the $\Delta$TOC trends (% per decade) over the century.

| Season | Region | $\Delta$TOC$^{2000,a}$ DU (%) | $\Delta$TOC$^{2100,a}$ DU (%) | $\Delta$TOC trends[b] (% per decade) |
|---|---|---|---|---|
| Annual mean | ~~NH MLs~~ | $-5.5 \pm 0.6$ ($-1.6 \pm 0.6$) | $-2.7 \pm 0.2$ ($-0.8 \pm 0.2$) | $0.10 \pm 0.03$ |
|  | Trop | $-2.1 \pm 0.3$ ($-0.8 \pm 0.2$) | $-1.5 \pm 0.1$ ($-0.6 \pm 0.1$) | $0.03 \pm 0.01$ |
|  | SH MLs | $-8.0 \pm 0.8$ ($-2.5 \pm 0.3$) | $-3.9 \pm 0.4$ ($-1.2 \pm 0.3$) | $0.15 \pm 0.04$ |
| DJF | NH MLs | $-6.2 \pm 0.1$ ($-1.7 \pm 0.6$) | $-3.2 \pm 0.3$ ($-0.8 \pm 0.2$) | $0.11 \pm 0.03$ |
|  | Trop | $-2.3 \pm 0.3$ ($-0.9 \pm 0.1$) | $-1.8 \pm 0.1$ ($-0.7 \pm 0.1$) | $0.02 \pm 0.01$ |
|  | SH MLs | $-6.9 \pm 0.6$ ($-2.4 \pm 0.4$) | $-4.5 \pm 0.2$ ($-1.5 \pm 0.1$) | $0.10 \pm 0.03$ |
| MAM | NH MLs | $-7.7 \pm 0.9$ ($-2.0 \pm 0.6$) | $-3.7 \pm 0.3$ ($-0.9 \pm 0.2$) | $0.14 \pm 0.04$ |
|  | Trop | $-2.1 \pm 0.2$ ($-0.8 \pm 0.2$) | $-1.5 \pm 0.1$ ($-0.6 \pm 0.1$) | $0.02 \pm 0.01$ |
|  | SH MLs | $-6.2 \pm 0.5$ ($-2.2 \pm 0.3$) | $-4.2 \pm 0.2$ ($-1.4 \pm 0.1$) | $0.11 \pm 0.03$ |
| JJA | NH MLs | $-4.1 \pm 0.6$ ($-1.3 \pm 0.2$) | $-2.3 \pm 0.2$ ($-0.7 \pm 0.1$) | $0.09 \pm 0.03$ |
|  | Trop | $-2.0 \pm 0.3$ ($-0.8 \pm 0.2$) | $-1.6 \pm 0.1$ ($-0.6 \pm 0.1$) | $0.02 \pm 0.01$ |
|  | SH MLs | $-8.0 \pm 0.9$ ($-2.5 \pm 0.2$) | $-3.2 \pm 0.4$ ($-0.9 \pm 0.1$) | $0.19 \pm 0.06$ |
| SON | NH MLs | $-3.8 \pm 0.4$ ($-1.2 \pm 0.1$) | $-1.6 \pm 0.1$ ($-0.5 \pm 0.1$) | $0.10 \pm 0.03$ |
|  | Trop | $-1.9 \pm 0.2$ ($-0.7 \pm 0.1$) | $-1.5 \pm 0.1$ ($-0.6 \pm 0.1$) | $0.01 \pm 0.01$ |
|  | SH MLs | $-10.0 \pm 1.2$ ($-3.0 \pm 0.8$) | $-5.7 \pm 0.4$ ($-1.6 \pm 0.8$) | $0.20 \pm 0.06$ |

[a] Absolute (DU) and relative (%; in brackets) TOC changes during the present-day period ($\Delta$TOC$^{2000}$) and the end of the 21st century ($\Delta$TOC$^{2100}$) were computed considering the mean ensemble of each experiment at mid-latitudes (~~SH MLs and NH MLs~~) and in the tropics (Trop). Annual and seasonal $\Delta$TOC errors were estimated from standard mean errors. [b] Trend errors were estimated as twice the statistical deviation stemming from the least-squares fit.

for the present day, which is in line with the projected $\Delta$TOC trends of 0.15 % and 0.10 % per decade for ~~SH MLs and NH MLs~~, respectively.

iii. Within the tropics, the inclusion of VSL$^{Br}$ leads to small impacts on the TOC compared to impacts observed at mid-latitudes. The TOC differences due to VSL$^{Br}$ are close to $-1.5$ DU ($< -1$ %) during practically the entire modelled period, with a maximum $\Delta$TOC$^{2000}$ reaching $-2.1$ DU ($< -1$ %). This is in line with a projected $\Delta$TOC trend of 0.03 % per decade. Moreover, even though VSL$^{Br}$ slightly worsens the agreement between the modelled TOC$^{LL}$ and observations, the minimum TOC$^{LL+VSL}$ is shifted to $\sim 5$ years earlier compared to TOC$^{LL}$ (i.e. towards where the inorganic-chlorine peak is located in the year 1995), which is in agreement with observations.

iv. Overall, our model results show much higher ozone depletion due to VSL$^{Br}$ at ~~southern and northern mid-latitudes~~ compared to the tropics.

Our results of the mid-latitude $\Delta$TOC due to VSL$^{Br}$ lie within the lower range of previous modelling studies over the 1960–2005 (Sinnhuber and Meul, 2015) and 1980–2005 (Feng et al., 2007) periods. ~~We expect to see the lower impacts modelled in this recent work, as for example the detailed treatment of VSL$^{Br}$ sources in the results of Sinnhuber and Meul (2015) for an additional Br$_y^{VSL}$ injection of~~

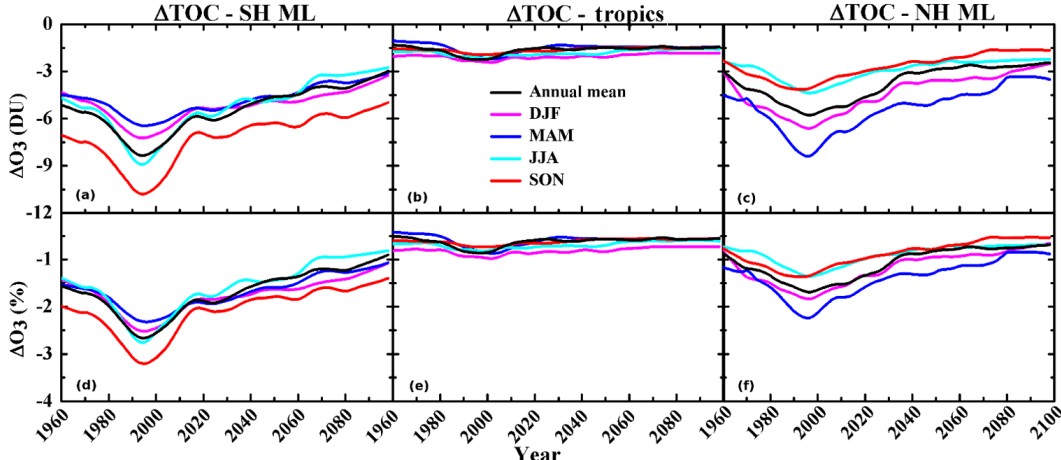

**Figure 4.** Temporal evolution of total ozone column difference ($\Delta$TOC) between the experiments. Panels **(a–c)** show the annual and seasonal mean absolute $\Delta$TOC (DU) at ~~SH MLs~~ in the tropics, and at ~~NH MLs~~ respectively, while panels **(d–f)** show the corresponding relative $\Delta$TOC (%).

~~6 ppt Br$_y$. The model configurations in Feng et al. (2007) included the direct supply of tropospheric Br$_y$ as a lower condition in addition to a VSL substance tracer.~~ Furthermore, our results are in accord with the future projection trends in Falk et al. (2017), although their analysis on annual mean ozone loss due to VSL$^{Br}$ only focused on the late 21st century (2075–2100).

Figure 4 shows the temporal evolution (1960–2100) of the annual and seasonal $\Delta$TOC between the experiments for SH MLs, the tropics, and NH MLs. In addition, an analysis of $\Delta$TOC as a function of latitude for the present day and the end of the 21st century is shown in Fig. 5. The inclusion of VSL$^{Br}$ leads to a continuous reduction in the TOC at all latitudes, with a larger ozone destruction efficiency moving from the tropics to the high latitudes. Moreover, note that when comparing the seasonal relative $\Delta$TOC between both periods, a statistically significant difference (here defined as $p < 0.05$) is only observed at mid-latitudes, as shown by the horizontal lines at the bottom of Fig. 5d. This highlights that the VSL$^{Br}$-driven ozone depletion efficiency changes significantly over the century even though the contribution from VSL$^{Br}$ ~~sources~~ to bromine stratospheric injection remains constant. Since gas-phase and heterogeneous inter-halogen reactions involving both bromine and chlorine play a fundamental role in stratospheric ozone loss (McElroy et al., 1986; Solomon, 1999 TS9; Salawitch et al., 2005), the ~~VSL$^{Br}$~~ efficiency in ozone depletion is primarily linked to the background inorganic-halogen abundance, which shows a continuous decline over the course of the 21st century~~, which is contemplated in~~ the A1 halogen emission scenario considered in this work. This is in line with the findings of Yang et al. (2014). Therefore, an additional stratospheric chlorine and bromine load from natural or anthropogenic substances not regulated by the Montreal Protocol will induce an increase in the VSL$^{Br}$ impacts modelled in this work on projected ozone loss trends throughout the current century.

The largest mid-latitude TOC differences between experiments occur during spring, when the maximum seasonal TOC levels are found, followed by the winter. The maximum springtime $\Delta$TOC$^{2000}$ reaches $-10$ DU ($\sim -3\%$) and $-7.7$ DU ($\sim -2\%$) for ~~SH MLs and NH MLs~~ respectively (see Table 1 and Fig. 4). In contrast, during summer and autumn, the maximum $\Delta$TOC$^{2000}$ ~~remains~~ below $-6.8$ ($\sim -2.4\%$) and $-4$ DU ($\sim -4.1\%$) for ~~SH MLs and NH MLs~~ respectively. This is in line with the ozone loss mediated by VSL$^{Br}$ within the southern polar cap reported by Fernandez et al. (2017), who showed a maximum ozone depletion of up to $-15$ DU ($\sim -14\%$ of TOC) during October at the beginning of the century. Moreover, as mentioned above, the projected seasonal VSL$^{Br}$ impact on the TOC significantly decreases towards the end of the century, with springtime $\Delta$TOC trends of up to $0.20\%$ and $0.14\%$ per decade for ~~SH MLs and NH MLs~~ respectively (see Table 1). In comparison, the projected $\Delta$TOC trends for summer and autumn reach approximately $0.10\%$ per decade at mid-latitudes. Note also that the changes in the magnitudes of these seasonal $\Delta$TOC trends are mainly attributed to a marked seasonal difference of ozone loss mediated by VSL$^{Br}$ during the period of highest background halogen abundance (i.e. present day). These seasonal differences are reduced towards the end of the 21st century due to the decline in long-lived inorganic-halogen abundance.

Within the tropics, no significant changes in the TOC loss due to VSL$^{Br}$ are projected, though the model captures a slightly major TOC depletion during the 1990s. Therefore, VSL$^{Br}$-driven ozone depletion is less sensitive to background halogen abundances, introducing an almost constant impact

**https://doi.org/10.5194/acp-20-1-2020**

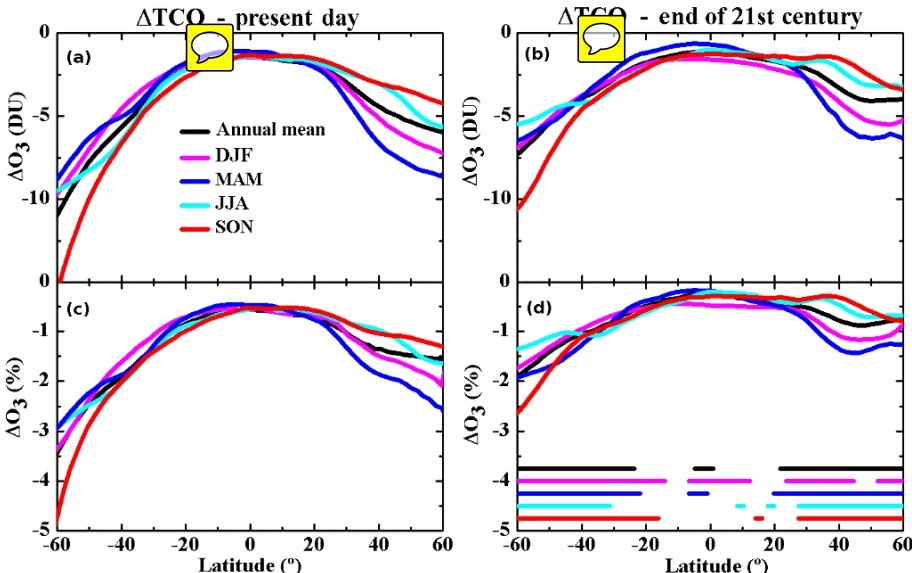

**Figure 5.** Latitude distributions of total ozone column difference ($\Delta$TOC) between the experiments. Panels **(a)** and **(b)** show the annual and seasonal mean absolute $\Delta$TOC (DU) for the present day and the end of 21st century, respectively, while panels **(c)** and **(d)** show the corresponding relative $\Delta$TOC (%). The corresponding horizontal lines shown in panel **(d)** indicate where the relative $\Delta$TOC between the present day and the end of the 21st century is statistically significant with a 95 % confidence interval using a two-tailed Student's $t$ test.

that is in line with its fixed emissions considered throughout the whole modelled period.

### 3.3 VSL$^{Br}$ impact on vertical ozone distributions

Timeline analysis of the annual mean stratospheric ozone difference between the experiments as a function of altitude at mid latitudes and in the tropics is presented in Fig. 6. The inclusion of VSL$^{Br}$ produces a reduction in ozone concentrations (i.e. $O_3(z)$ number densities) throughout most of the stratosphere during the whole modelled period. The largest $O_3(z)$ reductions occur during the present-day period in the lowermost mid-latitude stratosphere, reaching ozone differences ($\Delta O_3(z)$) of up to $-8$ % ($-0.25 \times 10^{12}$ molecules cm$^{-3}$) and $-5$ % ($-0.14 \times 10^{12}$ molecules cm$^{-3}$) for SH MLs and NH MLs, respectively. Above 50 hPa ($\sim 20$ km), $\Delta O_3(z)$ is practically constant at around $-1$ % for all latitudes, which is linked to the lesser role played by VSL$^{Br}$ compared to catalytic ozone loss by hydrogen ($HO_x = OH + HO_2$) and nitrogen ($NO_x = NO + NO_2$) oxides as well as by long-lived reactive halogens (Brasseur and Solomon, 2005; Salawitch et al., 2005; Müller, 2012).

Our modelled $\Delta O_3(z)$ latitudinal distribution is in agreement with those reported by Sinnhuber and Meul et al. (2015) during the 1980–2005 period, with a larger percentage of ozone loss deepening around the mid-latitude tropopause. In addition, $O_3(z)$ reduction due to VSL$^{Br}$ in the lowermost mid-latitude stratosphere is about 50 % smaller by the end of the 21st century compared to the present-day period (see Fig. 6d and e). This is in line with projected

local $\Delta O_3(z)$ trends of up to 1 % and 0.5 % per decade at SH MLs and NH MLs, respectively (see Fig. 6f). The largest $O_3(z)$ reductions are located precisely at the same altitudes and latitudes where the $ClO_x/Cl_y$ ratio peak is modelled within the lowermost stratosphere at SH MLs (see Fig. 2), which explains the hemispheric asymmetry in the VSL$^{Br}$ impacts. This highlights that the enhancement in local ozone depletion mediated by VSL$^{Br}$ is largest in the context of maximum background active-chlorine abundance since it provides an additional partner for chlorine species involved in the ozone-depleting inter-halogen chemical reactions (e.g. $ClO_x$–$BrO_{x\text{-Loss}}$; see Sect. 3.4). These results are in agreement with Yang et al. (2014), who determined that VSL$^{Br}$ has a significantly larger ozone impact in the presence of a high stratospheric chlorine background than a low chlorine background using a set of time slice sensitivity simulations with a variable and speculative VSL$^{Br}$ contribution on different background stratospheric $Cl_y$ abundances.

Figure 7 shows the seasonal zonal mean distribution of the $\Delta O_3(z)$ during the present-day period. The corresponding $\Delta O_3(z)$ trends (% per decade) over the century are shown in Fig. S6. The largest contributions to the annual mean ozone loss in the lowermost mid-latitude stratosphere correspond to spring, followed by the preceding winter and the posterior summer in each hemisphere, with springtime $\Delta O_3(z)$ reaching up to $-10$ % and $-7$ % for SH MLs and NH MLs, respectively (see Fig. 7b and d). This seasonal enhancement in VSL$^{Br}$-mediated local ozone depletion is influenced by the seasonal changes of the heterogeneous chlorine reactivation processes in lowermost stratospheric ice crystals since the

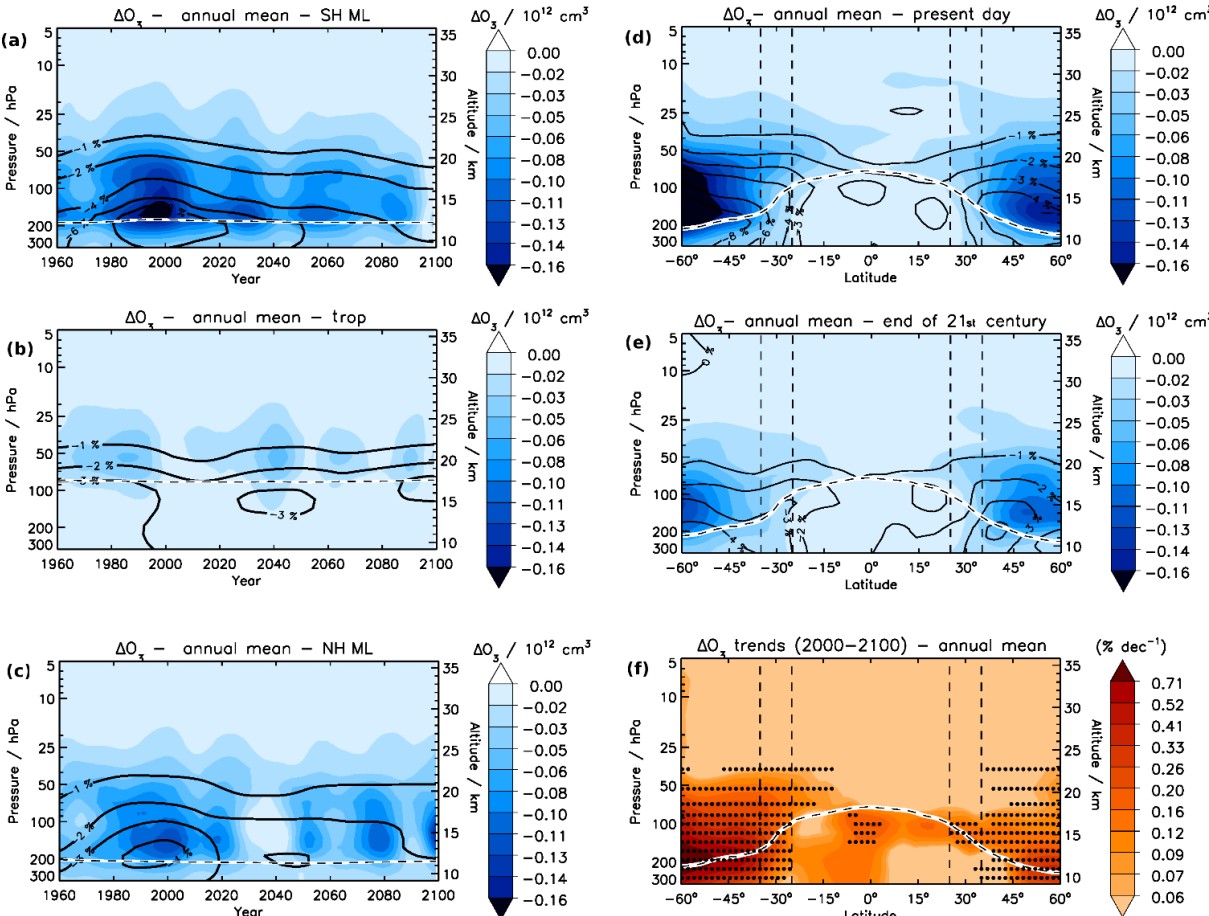

**Figure 6.** Temporal evolution of the annual mean ozone difference ($\Delta O_3$) between experiments as a function of altitude **(a)** at SH MLs, **(b)** in the tropics, and **(c)** at NH MLs. The right panels show the zonal mean $\Delta O_3$ distributions for the **(d)** present day and **(e)** the end of the 21st century as well as the **(f)** $\Delta O_3$ trends (% per decade) over the century. The absolute (colour scale) and relative (contour line) ozone differences were calculated considering the ozone number densities (i.e. molecules cm$^{-3}$) of each experiment. The masked regions in panel **(f)** indicate where the relative $\Delta O_3(z)$ between the present day and the end of the 21st century is statistically significant with a 95 % confidence interval using a two-tailed Student's $t$ test.

main heterogeneous bromine reactivation processes occur in sulfate aerosols and therefore lack seasonal changes (see Figs. S3 and S4). Consequently, since the seasonal changes in ozone loss are maximized during the present-day period, the largest local $\Delta O_3(z)$ trends are also projected for spring and winter, with springtime $\Delta O_3(z)$ trends reaching up to 1.6 % and 0.7 % per decade for SH MLs and NH MLs, respectively (see Fig. S6b and d).

Compared to the annual mean, springtime absolute $O_3(z)$ reductions along mid-latitudes are deepened over the first ~ 5 km above the tropopause during the present-day period, whereas these reductions are located at higher altitudes (between 2 and 5 km above the tropopause) and over a narrower geographic area, mainly at NH MLs, during summer and autumn. Therefore, VSL$^{Br}$ introduces seasonal changes not only in overall $O_3(z)$ reduction but also in vertical $O_3(z)$ distribution. Moreover, since seasonal VSL$^{Br}$ impacts persist until the end of the century, they could lead to significantly different seasonal perturbations of the radiative effects mediated by ozone. Future studies are needed to explore the potential VSL$^{Br}$-mediated seasonal effects on the atmosphere's radiative balance.

Near the tropical tropopause, the $O_3(z)$ reductions due to VSL$^{Br}$ are practically constant during the whole modelled period, with $\Delta O_3(z)$ of approximately $-2$ % ($-0.06 \times 10^{12}$ molecules cm$^{-3}$) at 50 hPa (see Fig. 6b). Even with an increase in the contribution of Br$_y^{VSL}$ to Br$_y^{LL+VSL}$ of up to 65 % by the end of century – due to the implementation of the Montreal Protocol (WMO, 2014, 2018) and the fixed stratospheric Br$_y^{VSL}$ injection – $\Delta O_3(z)$ changes in the tropical lowermost stratosphere are not significant over the century (see Fig. 6f). This is in line with the modelled impacts on the TOC. In contrast, at mid-latitudes, the seasonal VSL$^{Br}$ impacts on ozone significantly decrease over the cen-

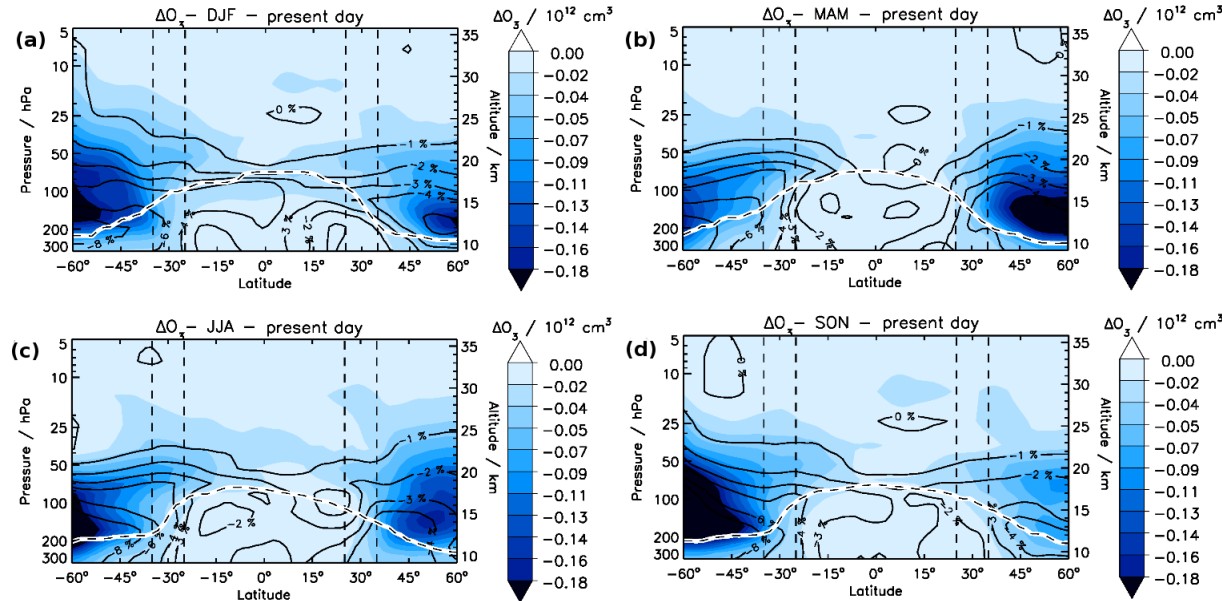

**Figure 7.** Seasonal zonal mean $\Delta O_3$ distributions during the present-day period.

tury between 50 and 300 hPa (see Figs. 6f and S6a–d) even though the contribution percentage from $Br_y^{VSL}$ to $Br_y^{LL+VSL}$ increases towards the end of the century. Moreover, the projected seasonal and annual $\Delta O_3(z)$ trends peak precisely at the same altitudes where the modelled relative $\Delta O_3(z)$ is significant (see Figs. 6f and S6a–d). These results highlight the clear dependence of VSL$^{Br}$ impacts on the temporal evolution of background halogen abundances and the halogens' heterogeneous reactivation processes, which in turn depend on temperature as well as the formation and distribution of ice crystals and stratospheric sulfate aerosol surfaces. In fact, the projected impacts on vertical ozone distribution at the end of the century are similar to those prevailing before 1980 for both SH MLs and NH MLs since the background halogen abundances are similar between these periods. Thus, potential future changes in the lowermost stratospheric temperature (therefore influencing ice crystal formation), geoengineering of climate via injection of stratospheric sulfate, and the formation and distribution of stratospheric aerosol (e.g. future volcanic eruptions) will directly influence the processes of heterogeneous halogen reactivation and consequently VSL$^{Br}$-driven ozone depletion efficiency (Tilmes et al., 2008, 2012; Banerjee et al., 2016; Klobas et al., 2017; Heckendorn et al., 2009).

### 3.4 Changes in the long-term seasonal evolution of halogen-driven ozone loss rate due to VSL$^{Br}$

In order to determine the drivers controlling the VSL$^{Br}$-driven seasonal impact on ozone in the lowermost stratosphere at mid-latitudes and in the tropics, we compute the odd-oxygen chemical loss from each ozone-depleting family considering the independent contributions of oxygen ($O_{x\text{-Loss}}$), hydrogen ($HO_{x\text{-Loss}}$), nitrogen ($NO_{x\text{-Loss}}$), and halogen (Halog$_{x\text{-Loss}}$; Brasseur and Solomon, 2005; Saiz-Lopez et al., 2014). Moreover, we discriminate between the individual contributions of pure bromine ($BrO_{x\text{-Loss}}$) and chlorine ($ClO_{x\text{-Loss}}$) cycles and the inter-halogen ($ClO_x - BrO_{x\text{-Loss}}$) cycle to the halogen family (Halog$_{x\text{-Loss}}$ = $BrO_{x\text{-Loss}} + ClO_{x\text{-Loss}} + ClO_x - BrO_{x\text{-Loss}}$). The odd-oxygen loss rate equations for the ozone-depleting families considered in this work are presented in Table S3.

Figure 8 shows the temporal evolution of the annual contribution percentage of each ozone-depleting family to the total ozone loss rate in the lowermost stratosphere at SH MLs ($\sim 120$ hPa) for both experiments as well as the seasonal evolution of $HO_{x\text{-Loss}}$ and Halog$_{x\text{-Loss}}$ contributions in each experiment. The corresponding figures for the lowermost stratosphere at NH MLs ($\sim 120$ hPa) and in the tropics ($\sim 50$ hPa) are shown in Figs. S7 and S8, respectively. Halog$_{x\text{-Loss}}$ represents the second most important contribution to the total ozone loss rate after $HO_{x\text{-Loss}}$ ($\sim 80\%$) at mid-latitudes in both experiments (see Figs. 8a and S7a), while within the tropics it represents the third family after $HO_{x\text{-Loss}}$ and $NO_{x\text{-Loss}}$ (see Fig. S8a). This partially explains the smaller modelled VSL$^{Br}$ impact on ozone within the tropics as Halog$_{x\text{-Loss}}^{LL+VSL}$ represents at most 10% of the total ozone destruction over this region. Moreover, the enhancement in the Halog$_{x\text{-Loss}}^{LL+VSL}$ contribution with respect to Halog$_{x\text{-Loss}}^{LL}$ by including VSL$^{Br}$ is mostly compensated by a decrease in the $HO_{x\text{-Loss}}^{LL+VSL}$ contribution at mid-latitudes. For example, for the present day, Halog$_{x\text{-Loss}}^{LL+VSL}$ shows an increase of $\sim 7\%$ with respect to Halog$_{x\text{-Loss}}^{LL}$ at SH MLs, which is compensated by a reduction of $\sim 6\%$ in

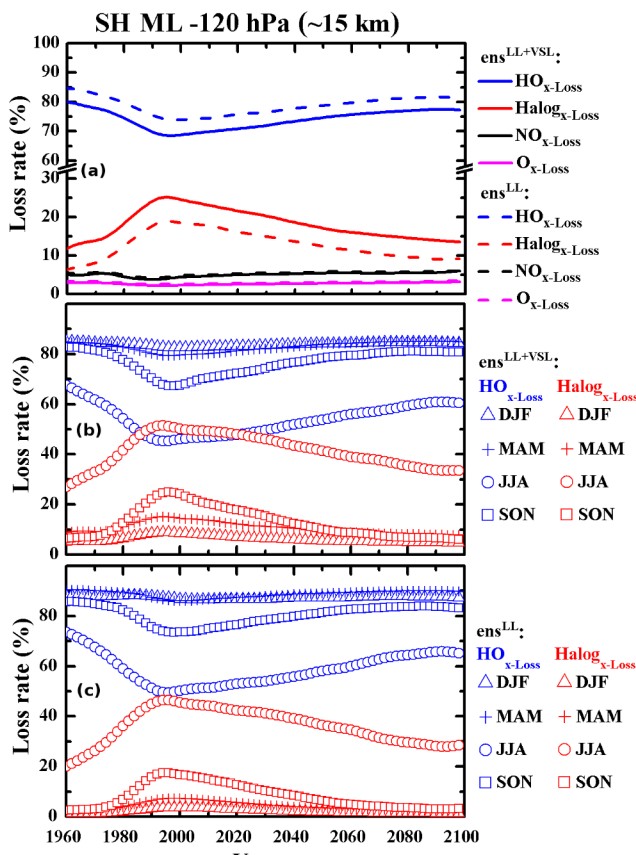

**Figure 8.** Temporal evolution of the contribution percentages from different ozone-depleting families ($O_{x\text{-Loss}}$, $NO_{x\text{-Loss}}$, $HO_{x\text{-Loss}}$, $Halog_{x\text{-Loss}}$) to the total odd-oxygen loss rate in the ~~lowermost stratosphere at SH MLs~~ (120 hPa). Panel **(a)** shows the annual mean contribution of each ozone-depleting family for the experiments with (solid line; $ens^{LL+VSL}$) and without (dashed line; $ens^{LL}$) $VSL^{Br}$ sources. Panels **(b)** and **(c)** show the seasonal mean contributions of both $HO_{x\text{-Loss}}$ and $Halog_{x\text{-Loss}}$ for the experiments with and without $VSL^{Br}$ sources, respectively.

$HO_{x\text{-Loss}}^{LL+VSL}$. In particular, the interaction between different catalytic ozone-depleting cycles involving OH and halogen radicals as well as the influence of reactive halogens in altering the $OH/HO_2$ ratio has been described elsewhere (Bloss et al., 2005; Saiz-Lopez and von Glasow, 2012).

Even though the additional seasonal ozone depletion due to $VSL^{Br}$ peaks during spring followed by winter at midlatitudes, the largest $Halog_{x\text{-Loss}}$ contribution to the total ozone loss rate occurs during winter in both experiments, presenting a much smaller contribution during the rest of the year (see Figs. 8b, c and S7b, c). Indeed, this marked seasonal behaviour prevails both during boreal winter at ~~NH MLs~~ and during the austral winter at ~~SH MLs~~. This is explained by considering the strong seasonal increase in active-chlorine abundance (driven by heterogeneous reactivation ~~in~~ ice crystals) as well as the winter decline in $HO_x$ abundance (driven by the decrease in solar radiation with respect to summer-

time). In fact, the enhancement in the $Halog_{x\text{-Loss}}^{LL+VSL}$ contribution due to $VSL^{Br}$ makes it the dominant ozone-depleting family during winter between 1985 and 2020 at ~~SH MLs~~, surpassing in importance the role played by the otherwise dominant $HO_{x\text{-Loss}}$. In contrast, no seasonal variability of $Halog_{x\text{-Loss}}$ contribution is modelled within the tropics in either experiment (see Fig. S8b and c), and thus the seasonal $VSL^{Br}$ impact on ozone remains approximately constant during the whole modelled period around the annual mean.

Figure 9 shows the long-term evolution of the contribution percentage of $BrO_{x\text{-Loss}}$, $ClO_{x\text{-Loss}}$, and $ClO_x-BrO_{x\text{-Loss}}$ to the $Halog_{x\text{-Loss}}$ as a function of months and years in the lowermost stratosphere at ~~SH MLs~~ (120 hPa) for both experiments. Equivalent results are presented for ~~NH MLs~~ and the tropics in Figs. S9 and S10, respectively. For any fixed year in both experiments, the $BrO_{x\text{-Loss}}$ contribution peaks during late summer and early autumn, while its minimum contribution occurs during the austral winter. In contrast, both the $ClO_{x\text{-Loss}}$ and $ClO_x-BrO_{x\text{-Loss}}$ reach their maximum contribution during winter and early spring. Consequently, the inclusion of $VSL^{Br}$ increases the $BrO_{x\text{-Loss}}^{LL+VSL}$ contribution in all seasons while also increasing the $ClO_x-BrO_{x\text{-Loss}}^{LL+VSL}$ contribution mainly in winter. This is at the expense of a decrease in the $ClO_{x\text{-Loss}}^{LL+VSL}$ contribution. In addition, the wintertime $ClO_x-BrO_{x\text{-Loss}}^{LL+VSL}$ contribution reaches up to 50 % between 1990 and 2020, decreasing towards the end of the century as $Cl_y^{LL}$ and $Br_y^{LL}$ abundances decrease within the A1 emission scenario. The decrease in both $ClO_{x\text{-Loss}}^{LL+VSL}$ and $ClO_x-BrO_{x\text{-Loss}}^{LL+VSL}$ over time is offset by an increase in the $BrO_{x\text{-Loss}}^{LL+VSL}$ contribution. Accordingly, the inclusion of $VSL^{Br}$ produces similar changes in the evolution of the contributions of each of the cycles that compose $Halog_{x\text{-Loss}}$ at ~~NH MLs~~, although the seasonal $BrO_{x\text{-Loss}}$ contribution is greater in both experiments compared to ~~SH MLs~~ during the entire modelled period (see Fig. S9).

In summary, the $Halog_{x\text{-Loss}}^{LL+VSL}$ contribution to the total ozone loss rate in summer and autumn is almost entirely dominated by $BrO_{x\text{-Loss}}^{LL+VSL}$ ~~, while in the winter months~~, when the seasonal $Halog_{x-Loss}^{LL+VSL}$ contribution to the total ozone loss rate is comparatively higher, a transition from the $ClO_x BrO_{x\text{-Loss}}^{LL+VSL}$ contribution ($\sim 50$ %) at the beginning of the century towards $BrO_{x\text{-Loss}}^{LL+VSL}$ at the end of the century is modelled. Thus, the halogen-driven ozone loss projected towards the end of the 21st century is mostly dominated by $BrO_{x\text{-Loss}}$, leading to a less marked seasonal $VSL^{Br}$ impact on both ~~vertical ozone~~ distribution and the TOC, in accordance with the modelled results. On the other hand, within the tropics, $BrO_{x\text{-Loss}}^{LL+VSL}$ dominates the seasonal $Halog_{x\text{-Loss}}^{LL+VSL}$ contribution for practically the entire modelled period (see Fig. S10), while $ClO_{x\text{-Loss}}^{LL+VSL}$ represents a maximum contribution of up to $\sim 35$ % during the end of the 20th century. The seasonal increase from $BrO_{x\text{-Loss}}^{LL}$ to $BrO_{x\text{-Loss}}^{LL+VSL}$ by including $VSL^{Br}$ is well marked and is prac-

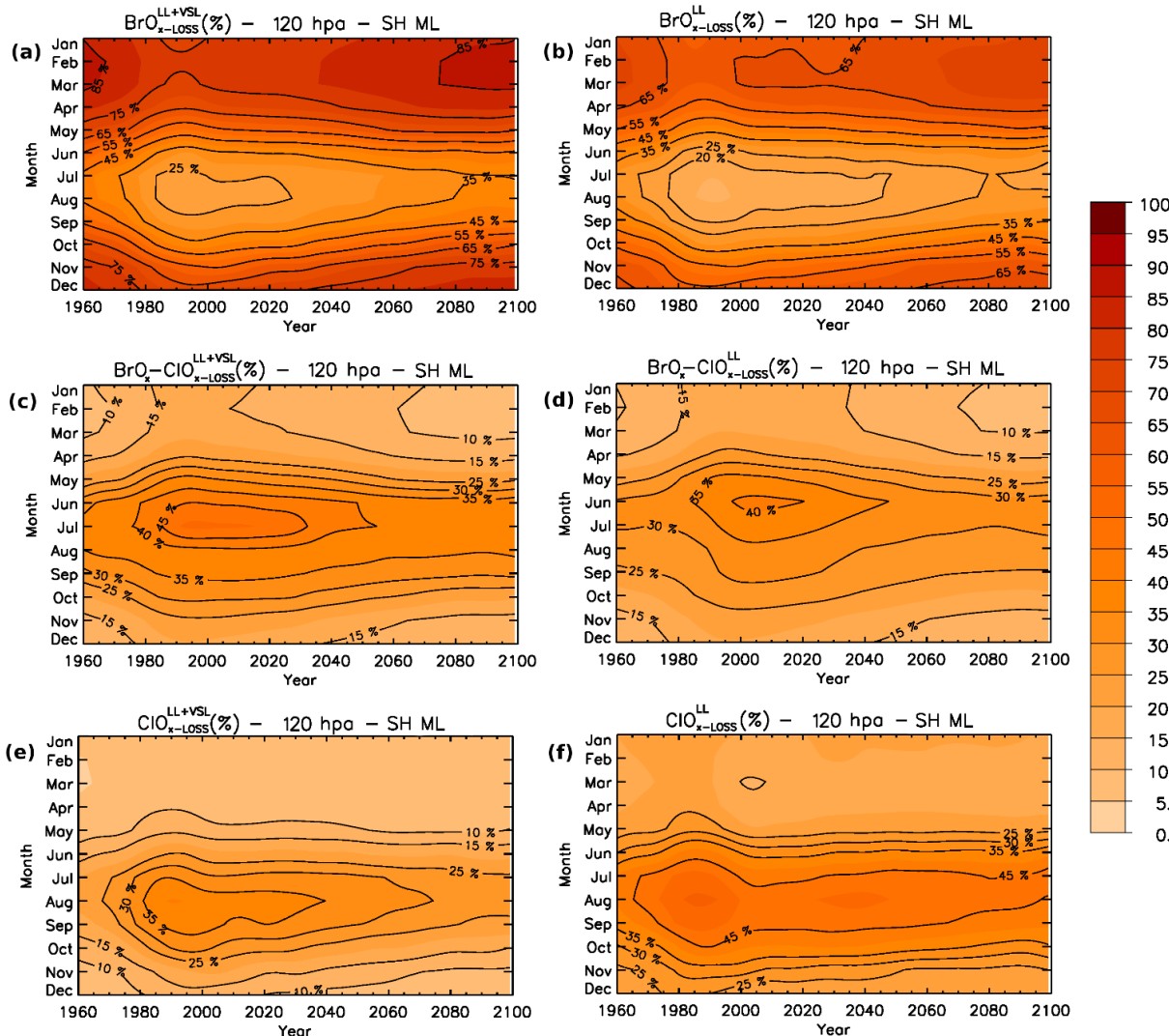

**Figure 9.** Evolution of the **(a, b)** $BrO_{x\text{-Loss}}$, **(c, d)** $ClO_x - BrO_{x\text{-Loss}}$, and **(e, f)** $ClO_{x\text{-Loss}}$ contribution percentages to $Halog_{x\text{-Loss}}$ as a function of the year and month in the lowermost stratosphere at SH MLs (120 hPa) for the experiments **(a, c, e)** with and **(b, d, f)** without $VSL^{Br}$ sources.

tically offset by a reduction in the $ClO_{x\text{-Loss}}^{LL+VSL}$ contribution. Furthermore, unlike mid-latitudes, $ClO_x - BrO_{x\text{-Loss}}^{LL+VSL}$ has a lower contribution to the tropical $Halog_{x\text{-Loss}}^{LL+VSL}$, showing that $VSL^{Br}$-driven topical ozone loss is independent of changes in the background halogen abundance throughout the modelled period.

## 4   Conclusions

This study has explored the impact of $VSL^{Br}$ substances on the temporal evolution of stratospheric ozone both in the late 20th century and in the course of the 21st century at mid-latitudes and in the tropics. The analysis compares two independent experiments, one of which considers only the anthropogenic $LL^{Cl}$ and $LL^{Br}$ source contributions, while

the other considers – in addition to long-lived sources – the contributions of $VSL^{Br}$ oceanic sources. We have evaluated annual and seasonal mean changes in vertical stratospheric ozone distribution and the TOC, and we have examined the projected ozone depletion ($\Delta O_3$ and $\Delta TOC$) trends over the century under a scenario in which both the relative contribution of $VSL^{Br}$ to total inorganic bromine CE10 and the background inorganic-chlorine abundances shift towards the future. Finally, we have also assessed the long-term seasonal contribution changes of halogen-mediated ozone loss ($Halog_{x\text{-Loss}}$) due to $VSL^{Br}$, focusing on the lowermost stratosphere and distinguishing the role of pure ($BrO_{x\text{-Loss}}$, $ClO_{x\text{-Loss}}$) and inter-halogen ($ClO_x - BrO_{x\text{-Loss}}$) cycles that compose $Halog_{x\text{-Loss}}$. The results presented here highlight the importance of considering natural sources of halocarbons

https://doi.org/10.5194/acp-20-1-2020

to determine the overall evolution of stratospheric ozone in the tropics and at mid-latitudes during the 21st century.

Our analysis shows that the inclusion of $VSL^{Br}$ results in a realistic stratospheric bromine loading that improves the agreement between the model and observations of the TOC at mid-latitudes, highlighting the need to consider these additional natural emissions to determine the overall evolution of ozone during the 21st century. Our results are in accordance with previous modelling studies that included natural bromocarbons and explored stratospheric ozone in the recent past (Salawitch et al., 2005; Feng et al., 2007; Sinnhuber et al., 2015) as well as by the end of the 21st century (Falk et al., 2017). However, unlike these previous works, we have explored seasonal ozone depletion arising from $VSL^{Br}$ with respect to the background stratospheric $LL^{Cl}$ and $LL^{Br}$ abundances between 1960 and 2100.

At mid-latitudes, $VSL^{Br}$ introduces a continuous reduction in the TOC during the entire modelled period, with the largest TOC depletion occurring during the period with the highest background halogen abundance ($\Delta TOC^{2000} = -8\,DU$, $-2.5\,\%$, for ~~SH MLs~~ and $-5.5\,DU$, $-1.6\,\%$, for ~~NH MLs~~). In turn, a significant decrease in the impact towards the end of the 21st century at mid-latitudes is projected, with $\Delta TOC$ trends of 0.15 % and 0.10 % per decade for ~~SH MLs and NH MLs~~, respectively. Even though the contribution of ~~$VSL^{Br}$~~ to total stratospheric bromine reaches almost 50 % by the end of the 21st century, our simulations show a statistically significant smaller $VSL^{Br}$ impact on lowermost stratospheric ozone as we move into the future. This result highlights that $VSL^{Br}$-driven ozone loss is closely linked to the temporal evolution of $LL^{Cl}$ and $LL^{Br}$ abundances regulated by the Montreal Protocol, as suggested by Yang et al. (2014). Along these same lines, Hossaini et al. (2015a) demonstrated that the chlorine load in the lowermost stratosphere has increased as a consequence of a recent and ongoing growth in emissions from anthropogenic $VSL^{Cl}$ sources (mainly $CH_2Cl_2$) not controlled by the Montreal Protocol. If this $VSL^{Cl}$ emissions trend continues into the future, additional studies will be required to quantify the increase in $VSL^{Br}$ impact modelled in this work on the projected ozone loss trends over the century considering the additional inorganic chlorine from $VSL^{Cl}$.

The largest TOC depletion associated with $VSL^{Br}$ corresponds to the spring months followed by winter, with maximum springtime $\Delta TOC^{2000}$ reaching $-10\,DU$ ($\sim -3\,\%$) and $-7.7\,DU$ ($\sim -2\,\%$) for ~~SH MLs and NH MLs~~, respectively. We find that the inclusion of $VSL^{Br}$ leads to seasonal changes in both local $O_3(z)$ reduction and its vertical $\Delta O_3(z)$ distribution within the lowermost stratosphere, which could result in different seasonal perturbations ~~of~~ the radiative effects mediated by ozone. This seasonal enhancement in $VSL^{Br}$-mediated local ozone loss is strongly influenced by the seasonal changes in ~~heterogeneous chlorine reactivation processes in~~ ice crystals (mainly by $ClONO_2$ and HCl recycling). In fact, the modelled additional increase in heterogeneous chlorine reactivation efficiency in the lower-

most stratosphere at ~~SH MLs~~ compared to ~~NH MLs~~ leads to hemispheric asymmetry in the $VSL^{Br}$-driven ozone reductions.

Even though the additional seasonal ozone depletion due to $VSL^{Br}$ peaks during spring followed by winter, the largest $Halog^{LL+VSL}_{x\text{-Loss}}$ contribution to the total ozone loss rate occurs during winter, with a much smaller contribution during all other seasons. In fact, the wintertime $Halog^{LL+VSL}_{x\text{-Loss}}$ contribution dominates the total ozone loss rate between 1985 and 2020 in the lowermost stratosphere at ~~SH MLs~~, with a $ClO_x\text{–}BrO^{LL+VSL}_{x\text{-Loss}}$ contribution to $Halog^{LL+VSL}_{x\text{-Loss}}$ reaching up to 50 %. Due to the expected decrease in the $LL^{Cl}$ and $LL^{Br}$ abundances, a transition from $ClO_x\text{–}BrO^{LL+VSL}_{x\text{-Loss}}$ to $BrO_{x\text{-Loss}}$ during the course of the 21st century is projected, leading to a less marked seasonal $VSL^{Br}$ impact towards the end of the century.

Within the tropics, although our model captures a slightly greater TOC depletion associated with $VSL^{Br}$ during the 1990s, no significant change is projected by the end of the 21st century. $Halog_{x\text{-Loss}}$ [CE11] represents the third family in the contribution to the total ozone loss rate after $HO_{x\text{-Loss}}$ and $NO_{x\text{-Loss}}$ in the tropical lowermost stratosphere, with an almost constant seasonal contribution throughout the modelled period. $BrO^{LL+VSL}_{x\text{-Loss}}$ practically dominates the contribution to $Halog^{LL+VSL}_{x\text{-Loss}}$, with a maximum $ClO_x\text{–}BrO^{LL+VSL}_{x\text{-Loss}}$ contribution of up to $\sim 30\,\%$ by the end of the 20th century, reflecting the low sensitivity of ~~$VSL^{Br}$~~ to background halogen abundances ~~, which allows it~~ to drive stratospheric ozone depletion.

*Data availability.* Computing resources, support, and data storage are provided and maintained by the Computational and Information System Laboratory from the National Center of Atmospheric Research (CISL, 2017). The code of the CAM-chem model can be downloaded from the website (https://www2.acom.ucar.edu/gcm/cam-chem 💬). Data that support the findings of this study can be downloaded from the AC2 web page (https://ac2.iqfr.csic.es/en/publications 💬).

*Supplement.* ~~The supplement related to this article is available online at: https://doi.org/10.5194/acp-20-1-2020-supplement.~~

*Author contributions.* RPF and ASL designed the experiments. RPF and JAB configured and ran all the simulations. JAB processed the results of all simulations, led the manuscript, and prepared the figures. RPF, ASL, FIS, and CAC provided additional feedback on geophysical processing and representation. JAB, RPF, FIS, CAC, JFL, and ASL [CE12] were involved in the discussion and iterations of the manuscript.

*Competing interests.* The authors declare that they have no conflict of interest.

*Acknowledgements.* ~~This study has been funded by the Agencia Nacional de Promoción Científica y Técnica (ANPCYT PICT-2016-0714, Argentina) and the European Research Council Executive Agency under the European Union's Horizon 2020 Research and Innovation programme (project "ERC-2016-COG 726349 CLIMAHAL").~~ ~~Rafael Pedro Fernandez and Javier Alejandro Barrera~~ would like to thank CONICET, UNCuyo (SeC-TyP M032/3853), and UTN (PID 4920-194/2018) for additional financial support. Alfonso Saiz-Lopez, Carlos Alberto Cuevas, and Fernando Iglesias-Suarez are supported by the Consejo Superior de Investigaciones Científicas (CSIC) of Spain. We are very thankful to ~~D.~~ Kinnison **TS12** for very helpful discussions and suggestions. We thank the two anonymous reviewers for their helpful comments that improved this paper.

*Financial support.* This research has been supported by the European Research Council Executive Agency under the European Union's Horizon 2020 Research and Innovation programme (grant no. ERC-2016-COG 726349 CLIMAHAL) ~~and~~ the Agencia Nacional de Promoción Científica y Técnica (grant no. ANPCYT PICT-2016-0714). **TS13**

*Review statement.* This paper was edited by Marc von Hobe and reviewed by two anonymous referees.

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

**Remarks from the language copy-editor**

CE1 Please confirm plural/singular usage throughout the text.
CE2 I have assumed at all instances of "and references therein" apply to all references in the citation, not just the last one. Please confirm throughout the text.
CE3 Please confirm the change.
CE4 Please confirm the change.
CE5 Please confirm the change.
CE6 Please confirm the change.
CE7 Is this inorganic bromine?
CE8 Is this active bromine?
CE9 Please confirm the change.
CE10 Do you mean inorganic-bromine abundances or just inorganic bromine?
CE11 Please confirm the change.
CE12 All authors must be mentioned explicitly in this section. Please confirm this change.

**Remarks from the typesetter**

TS1 The composition of Figs. 2–7 has been adjusted to our standards. This also includes language adjustments to Figs. 2–7.
TS2 Solomon et al. (1999) changed to Solomon (1999). Please confirm.
TS3 Sinnhuber et al. (2015) changed to Sinnhuber and Meul (2015). Pleas confirm.
TS4 Solomon et al. (1999) changed to Solomon (1999). Please confirm.
TS5 Solomon et al. (1999) changed to Solomon (1999). Please confirm.
TS6 Weber et al. (2013) changed to Weber et al. (2018). Please confirm.
TS7 Solomon et al. (1999) changed to Solomon (1999). Please confirm.
TS8 Solomon et al. (1999) changed to Solomon (1999). Please confirm.
TS9 Solomon et al. (1999) changed to Solomon (1999). Please confirm.
TS10 Please provide a reference list entry including creators, title, and date of last access.
TS11 Please provide a reference list entry including creators, title, and date of last access.
TS12 Please provide full first name for Kinnison.
TS13 Please note that there is a discrepancy between funding information provided by you in the acknowledgements and the funding information you indicated during manuscript registration, which we used to create this section. Please double-check your acknowledgements to see whether repeated information can be removed from the acknowledgements or changed accordingly. If further funders should be added to this section, please provide the funder names and the grant numbers. Thanks.
TS14 The reference of Douglass et al. (2008) is not mentioned in this paper. Please check.
TS15 Please provide place of publication.