# Peer review of "Seasonal impact of biogenic very short-lived bromine on lowermost stratospheric ozone between 60° N and 60° S during the 21st century"

_Atmospheric Chemistry and Physics, 2019_

## Referee Comment (RC1) · Anonymous Referee #1 · 27 Feb 2020

**Review ACP-2019-1091**

Seasonal impact of biogenic VSL bromine on the evolution of mid-latitude lowermost stratospheric ozone during the 21st century

**1 general comments**

`evaluating the overall quality of the discussion paper`

In this manuscript, the authors study the impact of brominated very short–lived substances $VSL^{Br}$ on the evolution of extratropical and tropical ozone in the lower stratosphere (LS).

The authors conducted two sensitive studies with and without an additional 5 ppt bromine source from VSLS in the stratosphere. An ensemble of three simulations for each scenario has been run from 1960s until the end of the 21st century with 10 years of spin-up from 1950–1960. Stratospheric halogen background loadings (e.g. CFCs, $CH_3Br$) follow the RCP 6.0 scenario. The model used was the CESM1. The model setup, forcing data, and experimental setup is described comprehensively.

This study complements and completes previous studies which focused on specific aspects or shorter/selective timespans and is therefore valuable in understanding the impact of $VSL^{Br}$ by many means. The authors especially:

- deduct the efficiency of ozone depletion efficiency under additional bromine load from VSL with respect to pure $CO_x$, $BrO_x$, and mixed $CO_x/BrO_x$ depletion and compare it to the efficiencies of $O_x$, $HO_x$, and $NO_x$ induced ozone losses;

- comprehensively study the impact of seasonal variation in the VLS source strength on ozone depletion in the UTLS, and

- attribute these to seasonal variation in chlorine heterogeneous reactions.

All studies in this manuscript are conducted for mid-latitudes in both hemispheres as well as the Tropics. Hence, the current title limiting the results to "mid-latitude" is misleading, because the authors study not exclusively mid-latitudes. I suggest to adjust the manuscript title to account for this.

The scientific content of the manuscript is sound, there are only a few minor remarks. The language is overall concise but needs some refinement where the statements are not entirely clear.

**2 specific comments**

`individual scientific questions/issues`

**Abstract**

- L17/L18/L22: *"extratropical"/"extra-polar"/"mid-latitudes"* Without further information, these terms may be interchangeable. If the authors, indeed, refer to the same region (latitude band), they should decide on ONE of the terms and use it consequently throughout the text. In case different regions are meant, the terms need further explanation since exact definitions vary from article to article. (Only in Section 2 a definition for "mid-latitudes" is given.)

**Section 1**

- P3L20: *"high-mid latitudes"* Same as above – how do the authors define this region?

- P3L37-38: *"[...] i.e. with an interactive ocean [...]"* How does this study benefit from using an interactive ocean if the authors consider fixed seasonal VSL emissions to reduce uncertainty (see Section 2)? This information might not belong to Section 1 and should at least be mentioned or discussed at the appropriate place in Section 2.

**Section 2**

- P4L32-P5L3: While the configuration of the model is very well described in this section, the authors do not discuss the consequences (if any) the low-top and low stratospheric resolution has on their analysis. They should elaborate on this.

- P5L3-4: *"The current CAM-Chem version includes a non-orographic gravity wave scheme [...], an internal computation of the quasi-biennial oscillation (QBO) [...]"* Which CAM-chem version do the authors refer to here? The current release is CAM6.0. Previously, they wrote about CESM1 with CAM4 and cited Tilmes et al. (2016). The referred article states regarding the QBO: "The limited vertical resolution of the FR model configurations does not allow for the generation of an internal QBO in CAM4-chem. Therefore, for the FR CCMI experiments, REFC1 and REFC2, the QBO is imposed in the model by relaxing equatorial zonal winds between 90 to 3 hPa to the observed interannual variability, following the approach by Matthes et al. (2010)." The authors should clarify the version which they are referring to and in case they meant CAM4, rephrase the sentence in a way that it is clear that the model does not generate an internal QBO.

- P5L6-7: *"[...] this model configuration uses a fully coupled Earth System Model approach, i.e. the ocean and sea ice are explicitly computed."* Doesn't this introduce additional uncertainties due to, e.g., radiative and temperature feedback? On the other hand, these ought be strongest in the Arctics which are not subject to this study. The authors should include a brief discussion on the advantages/disadvantages of this ESM approach on their study.

- P5L8-10: *"Two independent experiments (each of them with three individual ensemble members only differing in their 1950 initial condition) [...]"* For the sake of reproducible, how do these initial conditions differ? On climatological time-scales do initial conditions even matter?

**Section 3**

- P6L22-27: *"As the oceanic VSL$^{Br}$ emission [...]"* and the following *"Although the increase [...]"* The statement in the second sentence appears to be incomplete. With respect to the first statement (sentence), do the authors intent to say, that there are projected changes in atmospheric dynamics/chemistry (the ones they list) using the RCP 6.0 emission scenario suggesting a change in product gas injection (PGI), but they don't find these in their simulation? Or do they intent to say that the listed changes negate each other which is why they don't find any trend in PGI in their simulations? The authors should elaborate on this.

- P6L28: *"The validation of inorganic chlorine [...] clearly shows a good model-satellite agreement [...]"* The term "model-satellite agreement" may be considered as "lab-slang". One should rather write "good agreement between model and satellite" or "the model is in good agreement with satellite observations". Beside of this, it is not clear that the authors are referring to Fig. 1 as "validation".

- P6L32-34: *"The slight underestimation [...]"* If this is truly due to unaccounted anthropogenic chlorine sources, this means that the RCP 6.0 projections regarding future emissions of chlorinated species are erroneous. What are the implications on future ozone and climate? Based on their findings, the authors may discuss this in Section 4.

- P8L7-9: *"[...] introduces a continues reduction in TOC that exceeds the model ensembles variability between experiments [...]"* I might have missed it, but where is the "model ensembles variability" actually quantified in the manuscript?

- P8L18: *"[...] the largest model bias compared to SBUV [...]"* How does the model bias compare to the uncertainty in the SBUV data? If the authors cannot quantify this it should be at least discussed here.

- P8L20-21: *"[...] much larger ozone loss efficiency [...] within the mid-latitudes [...]"* Maybe point out that this is true in both hemispheres.

- P8L38-P9L2: *"[...] which shows a continuous decline in the course of the 21st century from its peak values observed during 2000 [...]"* As mentioned above, one has to be aware at this point, that decline this is due to the emission scenario for CFCs used in the model. As mentioned above and as the authors mentioned citing Hossini (2015b, 2017), if CFCs are not out-phased as demanded by the Montreal protocol or if new anthropogenic sources of halogenated substances occur in the future the projections will change. This should be, at least, mentioned in Section 4.

- P11L5-P12L22: This subsection's content is rather difficult to follow at times due to the abbreviations in use. Especially the terms, $\text{Halog}_{-\text{Loss}}$, $\text{BrO}_{\text{x}-\text{Loss}}$, and $\text{VSL}^{\text{Br}}$ are confusing. There seems to be no clear seperation between in the term $\text{Halog}_{-\text{Loss}}$ and $\text{BrO}_{\text{x}-\text{Loss}}$ with and without additional Br from short–lived substances. I will come back to this later on. The authors may elaborate on their terminology in this regard.

**Section 4**

In addition to the point mentioned above, since the results for the mid-latitude are substantially different from the ones for the Tropics, the authors may consider to separate these into different subsections to make it more clear.

**3 technical corrections**

purely technical corrections

**3.1 General**

- Although it is a matter of personal style/taste: There are many preceding conjunctions throughout the text which appear to be fillers. The authors may consider to remove those.

- The authors use $\underline{o}$ throughout the manuscript instead of $°$. This should be corrected for.

**3.2 Specific**

**Abstract**

- L20: *"[...] the inclusion of* $\text{VSL}^{\text{Br}}$ *result [...]"* result → results

- L33/L35 and others: *"The largest modelled [...] the spring [...]"* As long as not a specific spring or winter, etc. is meant, the prefix article is wrong.

- L36/L37-38: *"*$\text{Halog}_{-\text{Loss}}$*"* Just a remark: In the final typesetting this ought not be broken at the page's margin.

- L35-37: *"[...] the halogen-mediated ozone depletion [...] is more efficient than [...] respect to other seasons."* respect → with respect

**Section 1**

- P1L15-17: *"Owing to their short lifetimes, the impact of VSL on stratospheric ozone peaks at the extratropical lowermost stratosphere [...] an important atmospheric region [...]"* I suggest to insert "which is" → "[...] which is an important atmospheric region [...]" to make the sentence more readable.

- P1L34: imply → implies

- P3L2: *"[...] 3,290 ppt Cl and 19.6 ppt Br [...]"* Only a remark: It is rather uncommon to use ppt for such high concentrations (chlorine), but the authors probably intent to show the difference between chlorine and bromine concentrations in the most comprehensible way.

- P3L7: *"[...], understanding the role of natural VSL^{Br} sources is key for chemistry-climate projections."* Missing article in front of "key".

- P3L11/P4L22/P11L25: *"[...] has been reviewed elsewhere [...]"/"[...] has been given elsewhere [...]"/"[...] has been described elsewhere [...]"* Not sure if this is a proper citation style. Use "by" instead and have in-text citations?

- P4L5-9: *"The layout of the paper is as follows: Section 2 resumes the main characteristics [...]"* I assume "resume" is not the right word here. The authors probably meant "summarize".

**Section 2**

- P4L16-17: *"[...] with the exception that here we consider a constant and geographically distributed and seasonally-dependent oceanic emissions [...]"* Substitute "here" with "in this work" or something similar; emissions → emission.

- P4L18-19: *"It is worth noting that this emission inventory [...], which allows introducing a complete seasonal cycle on the emission strength [...]"* "It is worth noting that" can be dropped. As mentioned above, it merely serves as filler in this context; "allows introducing [...] on" → "allows to introduce [...] to".

- P4L22-23: *"Monthly and seasonally varying zonally averaged distributions lower boundary conditions of long-lived chlorine [...] were considered [...]"* I don't understand this sentence. Did the authors mean to write: "Monthly and seasonally varying zonally averaged distributions of long–lived chlorine [...] and bromine [...] were considered as lower boundary conditions [...]"?

- P4L22-28: *"Monthly and seasonally varying [...]"* This sentence is way to long. I suggest to break it up into two sentences. The natural breaking point would be L26 "[...], while [...]".

- P4L28-29: *"In order to avoid unnecessary uncertainties associated with the speculative evolution of VSLBr [...]"* The terms "unnecessary" and "speculative" are too judgmental in this context. You may rephrase to "[...] reduce uncertainties associated with the uncertain evolution [...]".

- P4L29-31: *"[...] we used a constant annual source strength for the whole modelled period [...]"* A "constant annual source strength" somehow implies constant emissions which is probably not what the authors intent to say based on the previous

sentences and the half-sentence in parenthesis which follows. It is clear that they treat the Ordoñez et al. emissions inventory as climatology, not projecting any future increase/decrease in emissions or changes in the seasonal cycle which they call "speculative". The authors should make this more clear.

- P4L32: *"The CAM-Chem configuration used here [...]"* here → in this work

- P5L21-22: *"[...], the zonal mean of the ensemble mean of each independent experiment (run$^{LL}$ and run$^{LL+VSL}$) [...]"* The usage of the term "run" seems to be ambiguous throughout the text. On P5L10/L13, a run refers to one single model integration or set-up but in this sentence and later on it seems to refer to the ensemble mean? The authors should elaborate on this.

- P5L33-37: Maybe the use of a list could increase the readability of this paragraph?

**Section 3**

- P6L5: *"In addition, Figure 1 also [...]"* Either "in addition" or "also" is unnecessary in this sentence, because they mean the same in this context. Please drop one or the other.

- P6L7: *"[...] the temporal evolution [...] show [...]"* show → shows

- P6L14/L46: present-time → present–day; present times → present–day

- P6L19-22: *"Although within the tropical lower stratosphere [...], the contribution [...]"* Consider to split this sentence at "the contribution".

- P6L34: *"Moreover, although [...]"* Filler, conjunctions can be removed.

- P7L18-20: *"A clear example is the fingerprints [...]"* This sentence is not clear. Do the authors mean to say: That the fingerprints of the Indian summer monsoon can be seen in the $ClO_x/Cl_y$ ratio at the given location (15 °N and 100 hPa)?; is → are

- P8L5-6: *"A list of interesting features can be observed [...]"* This phrase sounds colloquial. Maybe change to "Many interesting features [...]". Talking about lists, the author may consider to actually use a list for their points (i)-(iv). This would increase the readability of the text and content at this point.

- P8L33: *"[. . . ] the seasonal relatives [...]"* It is not clear, what the authors intent to say here, especially since "relatives" (= family members) is the wrong word. Do they mean the "relative difference in $\Delta$TOC for each season"? This should be clarified.

- P8L34: *"dotted"* This is probably not the right term in this context. There are actually no "dotted" lines in Fig. 5. All lines are solid, but some (in the lower part of the plot) display gaps which indicate the significance of the changes. The authors may consider changing their wording here.

- P9L16-21: *"In contrast to mid-latitudes [...] for the tropics [...]"* The results for the Tropics are the total opposite compared to mid–latitudes. To not confuse the results, it would increase the readability if this paragraph was to begin in a new line.

- P9L32-33: *"In contrast [...] is at least half-fold [...]"* "half-folded" is most likely not the proper term in this context. Did the authors mean that ozone loss due to $VSL^{Br}$ is reduced by $50\,\%$ by the end of the 21st century compared to present–day?

- P9L34: *"In agreement with our results, [...]"* This should be rephrased using "our results are in agreement with".

- P9L37: *"Interestingly, deepest O3(z) reductions [...]"* Drop "interestingly" if you don't mean to say that this is completely unexpected; here and similar at other places (e.g. P10L16): deepest $\rightarrow$ largest / greatest.

- P10L24-27: Some missing white spaces in front of or behind "-"; "Montreal Protoco" missing "l"

- P10L28: "This is line [...]" $\rightarrow$ "This is in line [...]"

- P11L13: *"[...] over the SG-ML lowermost stratosphere [...]"* over $\rightarrow$ in

- P11L30: seasonality increase $\rightarrow$ seasonal increase

- P11L37: *"crossed $ClO_x - BrO_{x-Loss}$"* Do you mean "mixed"?

**Section 4**

- P13L18: *"[...] winter. with maximum [...] reaching"* with $\rightarrow$ The; reaching $\rightarrow$ reaches

- P13L19-21: *"We find that the inclusion of $VSL^{Br}$ leads to seasonal changes on the overall depth and vertical distribution of ozone [...], which could result to [...]"* This sentence is not clear. By "overall depth", do you mean the TOC? changes on the $\rightarrow$ changes in the; results to $\rightarrow$ results in

- P13L30-33: *"[...] entirely dominated by $BrO_{x-Loss}$ [...]"* Also in Section 3 and elsewhere in the manuscript, the authors use the term $BrO_{x-Loss}$ to indistinguished between ozone depleted by $BrO_x$ from long–lived opposed $BrO_x$ from short–lived substances ($VSL^{Br}$)? This is somewhat confusing, as already mentioned above, since all organic bromine has to be transformed into inorganic bromine to deplete ozone.

**Figures**

The authors should, in general elaborate on:

- be more concise in captions. It is not always easy to see (without reading the whole caption) what each panel actually refers to since the relevant description is embedded in long sentences.

- choose a different color map for most of their contour plots since the "rainbow colors" imply visually a distinct divergence of data at the edge between blue and green/yellow. This may lead to unintended misinterpretation[1]. It is therefore depreciated, but unfortunately still widely used.

Fig. 1

 - With respect to the most common type of color-blindness, the authors may refrain from using green together with red in plots when there is no difference in line style or luminescence of the colors.

 Caption

 - L3-4: *"(a) global upper stratosphere (6.5 hPa) and lower stratosphere (50 hPa) (b) tropics [...]"* The way it is written, it looks as if "lower stratosphere [...]" is part of panel (a) instead of (b)-(d). The authors should make this more clear. Maybe use ";" instead of "and".

 - L5: *"show changes"* Wouldn't it be use the term "evolution" for consistency?

 - L7: *"The red triangles of all panels [...]"* You can drop "of all panels".

 - L9: dotted → dashed

Fig. 2

 See comment about color map.

Fig. 3

 Panel labels (a)-(i) move around quite a bit. Would it be possible to set them to lower left corner for each plot?

Fig. 6

 - The y-axes' ranges changes between the plots ((a,c) 300–4; (b) 100–4; (d,e,f) 300–5) hPa. Please make the range consistent in all of the plots.

 - Panel (f): See above comment on color map.

 Caption

 - L2: within of → within the

 - L3: shows → show

Fig. 8

- The axes' lines are far thicker than the actual markers or lines symbolizing the data which is distracting. Partly due to this the violet line and especially the black line at the very bottom in panel (a) are easily missed. Please reduce the axes thickness and think about to change the color for $O_{x-Loss}$.

- The legends are overlapping with the content and should be placed outside of the plot area (right hand side).

- The plots (b) and (c) are extremely busy and not well explained in the caption. Why are data for JJA displayed using dashed lines in contrast to the thin, solid line/marker combination used for the remaining "seasons"? You should rather use either four different line styles or markers. Please make this consistent.

- Is it possible to show the actual ensemble standard deviation in this plot? It is mentioned in the text but nowhere quantified.

  Caption

- L4-6: *"The panels (b) and (c) show the seasonal mean contributions [...]"* This is not concise. Already is it difficult to spot LL and LL+VSL in the plots' legends, but the way this is presented in the caption is not making it any easier to interpret this plot. The actual information in what (b) and (c) differ comes in the very end of a long sentence. A better approach would be, e.g. "Seasonal mean contributions of [...] families for the different experiments (b) run$^{LL+VLS}$ and (c) run$^{LL}$."

Fig. 9

- See comment about color map above.

- The color bar indicates an open interval in both direction (blue and red triangles). If the data actually is confined in a closed interval between 0 and 100, the triangles should not to be used.

- Can you show or quantify the magnitude of difference between the percental contribution of $BrO_{x-Loss}$, $BrO_x - Cl_{x-Loss}$, and $ClO_{x-Loss}$ in both, run$^{LL}$ and run$^{LL+VSL}$? One cannot draw this information of a plot with color levels of widths 10 % especially for panels (c,d). (Even worse in Fig. S9).

Supplementary figures

All comments above apply to similar figures displayed in the supplement.

**References**

[1] D. Borland and R. M. Taylor Ii, *LATEX: Rainbow Color Map (Still) Considered Harmful*, IEEE Computer Graphics and Applications, vol. 27, no. 2, pp. 14-17, March-April 2007.

---

## Referee Comment (RC2) · Anonymous Referee #2 · 30 Mar 2020

**Review of 'Seasonal impact of biogenic VSL bromine on the evolution of mid-latitude lowermost stratospheric ozone during the 21st century' by Javer A. Barrera et al.**

The manuscript presents a modelling study of the impact of very short-lived halogenated compounds (VSL$^{Br}$) on stratospheric ozone under current and future conditions. In particular, seasonal changes of the impact VSL$^{Br}$ on midlatitude ozone and their link to heterogeneous chlorine reactivation are discussed. While many findings of this study are known from existing publications, the seasonality of the changes presented here is a unique and new result. The manuscript is well written, and the study is of interest to the readership of ACP. I recommend publication after addressing the following comments.

General comments

1) The CAM-Chem version used here does not include the full stratosphere and has a relatively low lid when compared with state-of-the-art chemistry climate models. Therefore, the full Brewer-Dobson circulation path is not included in the model domain. It is not clear how modelled future changes of the circulation and stratospheric temperatures are impacted by the low model lid and a discussion of these facts should be included in the manuscript.

2) A critical discussion of the how the projected changes of lowermost stratospheric temperatures impact the role of VSL$^{Br}$ driven ozone loss is needed. Both points (1 and 2) should be used to demonstrate that the model version used here offers an appropriate model set up.

3) The discussion of the VSL$^{Br}$ emission parametrization should make clear that the seasonality of the emissions is not well known due to missing process understanding and sparse observational data.

4) Why are chlorinated very short-lived substances not included in the halogen budget? Their current and potential future emissions could play an important role and it would be interesting to see how this impacts the relative role of VSL$^{Br}$.

5) Please explain why the polar latitudes are not analysed.

Specific comments

1) Page 5, line 6-7. Doe this refer to ocean dynamics or also to ocean biogeochemistry?

2) Figure 1. Legend is too small, and therefore it was very hard to read the figure.

3) Page 6, line 32-33. It is not clear why not including the short-lived chlorinated species would cause an offset to the observed Cly only in the tropics. Once they contributed to the inorganic chlorine budget, they should play a role at all latitudes.

4) Page 8, line 13. Here and in several other places, a consistency between different values is cited that is not necessarily clear to the reader. Why is the number of the VSL$^{Br}$ impact being 50% at the end of the century consistent with projected delta TOC changes of 10 to 15%?

5) Page 8, line 19. Why is the minimum TOC shifted to 5 years earlier when including a constant VSL$^{Br}$ term?
6) Page 8, line 34 and other places. What is the term VSL$^{Br}$ driven ozone loss efficiency referring to? Is this describing the total impact of VSL$^{Br}$ on ozone or some relative terms (% ozone loss)?
7) Page 9, line 1. Something is missing in the sentence.
8) Page 11, line 20 and 21. This is not clear and might need some more explanation. First of all, Figure 8 (which has again a way too small legend) seems to suggest that the Ox and NOx terms also increase. Is this increase maybe similar to the one of the HOx term relative to the total contribution? If not, it would be worthwhile explaining why the VSL impact gets compensated by the HOx term alone.

---

## Author Comment (AC1) · 7 May 2020

**Seasonal impact of biogenic very short-lived bromine on lowermost stratospheric ozone between 60º N and 60º S during the 21st century**

5    **Barrera Javier A. et al.**

**General Answer**

We are very grateful to Anonymous Reviewers No. 1 and 2 for their constructive comments and suggestions, which helped us to improve the draft. In the present revised version we have

10    taken into account all the reviewer's comments, including the adjustment of manuscript title, clarifying descriptions of model configurations, rephrasing of misleading implications, updating of some confusing terminologies and the revision of the quality/format of all figures as well as their corresponding captions. We have also introduced two tables and an extra figure to the supplementary material as a complement to the analysis presented in the main text.

15    To facilitate the reading, the original comments made by the reviewers have been copy-pasted here using **bold font**, while our answers are given in regular font. Additionally, we have copied into this response letter the current changes made to the original draft, using a *blue (corrected text)* and/or *italic (original text)* font type.

20    *******************************************************************************

**1   Anonymous Reviewer #1**
* * *
**1.1  General comments**

**In this manuscript, the authors study the impact of brominated very short–lived sub-**
25    **stances VSL$^{Br}$ on the evolution of extratropical and tropical ozone in the lower strato-**
**sphere (LS).**

**The authors conducted two sensitive studies with and without an additional 5 ppt bromine source from VSLS in the stratosphere. An ensemble of three simulations for each scenario has been run from 1960s until the end of the 21st century with 10 years of spin-up from**
30    **1950–1960. Stratospheric halogen background loadings (e.g. CFCs, CH$_3$Br) follow the RCP 6.0 scenario. The model used was the CESM1. The model setup, forcing data, and experimental setup is described comprehensively.**

**This study complements and completes previous studies which focused on specific aspects or shorter/selective timespans and is therefore valuable in understanding the impact of**
35    **VSL$^{Br}$ by many means. The authors especially:**

- **deduct the efficiency of ozone depletion efficiency under additional bromine load from VSL with respect to pure ClO$_x$, BrO$_x$, and mixed ClO$_x$/BrO$_x$ depletion and compare it to the efficiencies of O$_x$, HO$_x$, and NO$_x$ induced ozone losses;**

- **comprehensively study the impact of seasonal variation in the VLS source strength**
40    **on ozone depletion in the UTLS, and**

– **attribute these to seasonal variation in chlorine heterogeneous reactions.**

**All studies in this manuscript are conducted for mid-latitudes in both hemispheres as well as the Tropics. Hence, the current title limiting the results to "mid-latitude" is misleading, because the authors study not exclusively mid-latitudes. I suggest to adjust the manuscript title to account for this.**

**The scientific content of the manuscript is sound, there are only a few minor remarks. The language is overall concise but needs some refinement where the statements are not entirely clear.**

**1.2 Specific comments**

**Abstract**

**L17/L18/L22: "extratropical"/"extra-polar"/"mid-latitudes" Without further information, these terms may be interchangeable. If the authors, indeed, refer to the same region (latitude band), they should decide on ONE of the terms and use it consequently throughout the text. In case different regions are meant, the terms need further explanation since exact definitions vary from article to article. (Only in Section 2 a definition for "mid-latitudes" is given.)**

We have modified the terms "extratropical/extra-polar" of original manuscript by the following terms:

P1L16: "extratropical lowermost stratosphere" by "lowermost stratosphere"

P1L18: "extra-polar regions" by "within the tropics and mid-latitudes"

P2L17: "extratopical lowermost stratosphere" by "lowermost stratosphere"

P4L13 "extra-polar stratospheric ozone" by "in the tropics and mid-latitudes"

P14L30: "extra-polar stratospheric ozone" by "in the tropics and mid-latitudes"

The mid-latitudes and tropics regions have been defined in the section 2.

**Section 1**

**P3L20: "high-mid latitudes" Same as above – how do the authors define this region?**

We have specified the latitudinal band (40–75º N) that comprises the "northern high-mid latitudes" region according to the work of Spang et al, (2015), as follows:

P3L32: [...] *The occurrence of cirrus clouds was reported in the lowermost stratosphere of northern high-mid latitudes (40–75º N) (Spang et al., 2015)* [...]

**P3L37-38: "[...] i.e. with an interactive ocean [...]" How does this study benefit from using an interactive ocean if the authors consider fixed seasonal VSL emissions to reduce uncertainty (see Section 2)? This information might not belong to Section 1 and should at least be mentioned or discussed at the appropriate place in Section 2.**

We agree with the reviewer that explicitly mentioning the interactive ocean coupling within the Introduction Section can introduce confusion. Thus, we moved "i.e. with an interactive ocean" from Section 1 to Section 2.

We also thanks for highlighting the importance of making clear why we used an interactive-ocean coupling if we forced our model with annually-cycled $VSL^{Br}$ fluxes throughout the whole modeling period:

1-The interactive ocean coupling is part of the CCMI recommendation for REFC2 simulations, and we followed that procedure as CAM-Chem had an excellent tropospheric and stratospheric performance using that configuration (see Tilmes et al., 2016 and references therein);

2-The rationale for considering annually-cycled $VSL^{Br}$ fluxes (as clearly captured by the Reviewer) was to reduce the uncertainties associated with the uncertain evolution of both biogeochemical production and oceanic emissions of $VSL^{Br}$ into the future.

To make these points clear, we have removed lines P4L28-31 of the original manuscript and included the following lines in a new paragraph in Section 2:

P5L29-35: *Even though the interactive ocean coupling would have allowed us to compute the evolution of $VSL^{Br}$ ocean source emissions throughout the modelled period, we still lack a considerable understanding of the seasonal processes that dominate $VSL^{Br}$ ocean emissions in both present and future time, which is combined with a limited observation data set on VSL bromocarbons (WMO, 2014; WMO 2018). Therefore, in this work, we forced the model with annually-cycled $VSL^{Br}$ fluxes, replicating the Ordoñez et al. emissions inventory for all years between 1950 and 2100. This procedure aims to reduce the uncertainties associated with the uncertain evolution of both biogeochemical production and oceanic emissions of $VSL^{Br}$ into the future (Lennartz et al.; 2015; Ziska et al., 2017).*

**Section 2**

**P4L32-P5L3: While the configuration of the model is very well described in this section, the authors do not discuss the consequences (if any) the low-top and low stratospheric resolution has on their analysis. They should elaborate on this.**

Our full $VSL^{Br}$ state-of-the-art chemical scheme has only been implemented in CAM-Chem, and not in higher top models such as WACCM. Thus, we cannot perform this type of evaluation using a high-top model.

In particular, the region of maximum gravity-wave dragging, as well as most of the Brewer-Dobson circulation, is located above CAM-Chem model top. Therefore to reproduce the overall stratospheric circulation, the integrated momentum that would have been deposited above the model top is parameterized and deposited in the top layer of the model as an upper boundary condition (Lamarque et al., 2008), resulting in a consistent description of the stratospheric dynamics.

Even when CAM-Chem has a relatively low model top (~40 km), the stratospheric performance of CAM-Chem has been extensively inter-compared against the widely used WACCM configuration of CESM (i.e., considering 56 levels up to ~80 km), showing an excellent agreement in reproducing the large-scale changes of the stratospheric chemical composition, radiation and climate. For example: Fig. 2 in Lamarque et al. (2008), shows an

inter-comparison of photolysis rates of active radiative species throughout the whole vertical profile, while Fig. S1 in Fernandez et al. (2017) compares the climatic evolution of the stratospheric Age of Air and Total Ozone Column evolution through the 21st century. It should be noted that CAM-Chem was part of CCMVal-2 (Eyring et el., 2010) and CMIP5 (Eyring et al., 2013) inter-comparison projects, presenting ozone evolution trends and return dates estimations that lies in the middle of the multi-model range. To make this clear, we have included the following lines in the "Methods" Section of the new version of the manuscript:

P5L11-14: [...] *To ensure a consistent dynamical description of the stratosphere within the relatively low model top of CAM-Chem (i.e., gravity-wave dragging and the Brewer-Dobson circulation), the integrated momentum that would have been deposited above the model top is specified by an upper boundary condition (Lamarque et al., 2008). A similar procedure is applied to the altitude-dependent photolysis rate computations, which include an upper boundary condition that considers the ozone column fraction prevailing above the model top.* [...]

P5L21-24: [...] *Equivalent CAM-Chem configurations, as the one implemented in this work, have been used for the CCMVal-2 (Chemistry-Climate Model Validation 2) and CMIP5 (Coupled Model Intercomparison Project Phase 5) activities, in order to represent trends in the ozone evolution and to estimate the return dates, which lie in the middle of the multi-model range (Eyring et al., 2010; 2013)* [...]

**P5L3-4: "The current CAM-Chem version includes a non-orographic gravity wave scheme [...], an internal computation of the quasi-biennial oscillation (QBO) [...]" Which CAM-chem version do the authors refer to here? The current release is CAM6.0. Previously, they wrote about CESM1 with CAM4 and cited Tilmes et al. (2016). The referred article states regarding the QBO: "The limited vertical resolution of the FR model configurations does not allow for the generation of an internal QBO in CAM4-chem. Therefore, for the FR CCMI experiments, REFC1 and REFC2, the QBO is imposed in the model by relaxing equatorial zonal winds between 90 to 3 hPa to the observed interannual variability, following the approach by Matthes et al. (2010)." The authors should clarify the version which they are referring to and in case they meant CAM4, rephrase the sentence in a way that it is clear that the model does not generate an internal QBO.**

The reviewer is right. We have changed the sentence in the original manuscript following Tilmes et al. (2016), including the model version used in this work and how it represents the oscilación cuasi bianual (QBO). We have modified the text as follows:

P5L16-20: [...] *The CAM4-Chem version used in this work* includes a non-orographic gravity wave scheme based on the inertia-gravity wave parameterization and an observational-based implementation of stratospheric aerosol and surface area density. *The quasi-biennial oscillation is imposed in the model by relaxing equatorial zonal winds between 90 to 3 hPa to the observed interannual variability (see Tilmes et al., 2016 for more details)*

**P5L6-7: "[...] this model configuration uses a fully coupled Earth System Model approach, i.e. the ocean and sea ice are explicitly computed." Doesn't this introduce additional uncertainties due to, e.g., radiative and temperature feedback? On the other hand, these ought be strongest in the Arctics which are not subject to this study. The authors should include a brief discussion on the advantages/disadvantages of this ESM approach on their study.**

The interactive ocean coupling is recommended for climatic REFC2 1950-2100 simulations to make the representation of climate change in the models more physically self-consistent. To isolate the "chemical impact" of $VSL^{Br}$ in the present time atmosphere, we've already performed perpetual 2000 sensitivity simulations (with/without VSL sources) forced with identical meteorological fields, and determine the changes in inorganic bromine burden, abundance and impacts (Fernandez et al., 2014). However, to determine the climatic impact of $VSL^{Br}$ sources into an evolving stratosphere, we need to explicitly consider the radiative and temperature feedbacks. Indeed, the REFC2 CCMI recommendation suggests performing coupled sea-ice and ocean simulations for the 1950-2100 periods (see Table 2 in Eyring et al., 2013, where it is explicitly mentioned that "*Development should continue towards comprehensive troposphere-stratosphere CCMs, which include an interactive ocean, tropospheric chemistry, a naturally occurring QBO, spectrally resolved solar irradiance, and a fully resolved stratosphere (P49)*"). We truly believe that the current CAM-Chem REFC2 configuration considers all these issues and have been already widely validated in both the troposphere and stratosphere (see previous answers). To highlight all of this, we have incorporated the following sentences in the corrected manuscript:

P5L25-28: [...] *This implies that instead of isolating the chemical impact mediated by $VSL^{Br}$ sources on the climatological ozone budget (as would be the case using a specified dynamic approach), the chemical interaction between VSL-bromine and ozone in the lowermost stratosphere is affected by dynamic feedbacks (i.e. temperature, radiation, etc.).* [...]

**P5L8-10: "Two independent experiments (each of them with three individual ensemble members only differing in their 1950 initial condition) [...]" For the sake of reproducible, how do these initial conditions differ? On climatological time-scales do initial conditions even matter?**

Individual climatic (1950-2100) simulations, even when computed with the same model and configuration, have been show to present a non-negligible variability depending on the initial conditions used.

Indeed, "Due to the inertia of the coupled atmosphere ocean system, such integrations should be started from equilibrated control simulations for preindustrial conditions, as is standard for the 20th century integrations in CMIP5" (Eyring et al., 2013).

To assure our results reproducibility and highlight the importance of considering multiple ensemble experiments, we have included in the supplement the ozone results for each of the individual simulations of the each experiment and modified the text as follows:

P5L37-P6L2: *Two ensembles of independent modelling experiments were performed, each with three individual simulations from 1960 to 2100, considering approximately 10-year of spin-up to allow stabilization of the stratospheric circulation. The individual simulations 001,002 and 003 started in 1950, 1951 and 1952, respectively, with initial conditions taken from the equivalent years of the CESM1-WACCM (Whole Atmosphere Community Climate Model) 20th Century ensemble for CMIP5 (Marsh et al., 2013)* [...]

P6L6-7: *The mean ensemble of each experiment (i.e. with and without the $VSL^{Br}$ sources) was calculated as the average of the three individual simulations.* [...]

P9L19-20: [...] *Equivalent results for the temporal evolution of the individual simulations of each experiment, are shown in Fig. S5* [...]

**Section 3**

**P6L22-27: "As the oceanic VSL^{Br} emission [...]" and the following "Although the increase [...]" The statement in the second sentence appears to be incomplete. With respect to the first statement (sentence), do the authors intent to say, that there are projected changes in atmospheric dynamics/chemistry (the ones they list) using the RCP 6.0 emission scenario suggesting a change in product gas injection (PGI), but they don't find these in their simulation? Or do they intent to say that the listed changes negate each other which is why they don't find any trend in PGI in their simulations? The authors should elaborate on this.**

As indicated by the reviewer, we have rephrased the sentences of the lines P5L22-27 of original manuscript to make this clearer, as shown below:

P7L31-P8L4: [...] *Falk et al. (2017) showed an increase of VSL^{Br} of about 0.5 ppt in the tropical lowermost stratosphere by the end of the century compared to present-day, under the RCP6.0 scenario and assuming constant VSL^{Br} fluxes, following scenario five of Warwick et al. (2006). This increase in the stratospheric VSL^{Br} attributed to enhanced vertical transport in the tropics, is counteracted by a decrease of the $Br_y^{VSL}$ injected into the stratosphere, so that the total stratospheric VSL-bromine remains unchanged between the two periods. In this work, the stratospheric injection of VSL^{Br} and $Br_y^{VSL}$ between 2000 and 2100 remain unchanged. Based on the limited observational data, it is uncertain to draw a conclusive statement whether any trend in the stratospheric injection of organic and inorganic VSL-bromine would be expected. Nevertheless, our stratospheric injection of VSL^{Br} (~2 ppt) and $Br_y^{VSL}$ (~3 ppt) at present-day is in perfect accordance with that reported in the latest WMO (2018). Further studies are needed to evaluate the uncertain evolution of the stratospheric injection of organic and inorganic species from VSL^{Br} sources throughout the 21^{st} century.*

**P6L28: "The validation of inorganic chlorine [...] clearly shows a good model-satellite agreement [...]" The term "model-satellite agreement" may be considered as "lab-slang". One should rather write "good agreement between model and satellite" or "the model is in good agreement with satellite observations". Beside of this, it is not clear that the authors are referring to Fig. 1 as "validation".**

We have modified the "good model-satellite agreement" term used throughout the original manuscript to "good agreement between the model and the observations". We have included a Table (S1) in the supplemental material indicating the bias, normalized mean bias and normalized mean error as a measure to evaluate the agreement between modelled $Cl_y^{LL}$ and MLS (HCl+ClO) observations.

We have modified the text as follows:

P7L13-15: The values of bias, normalized mean bias (NMB) and normalized mean error (NME) to evaluate the agreement between the modelled $Cl_y$ and observations for each of the regions under study, are shown in Table S1. [...]

P8L5-9: *The modelled inorganic chlorine abundance shows a good agreement with the (HCl + ClO) MLS observations for the 2005-2015 period, mainly in the upper stratosphere where most chlorine has already been photochemically converted to $Cl_y^{LL}$. For example, the normal mean error of the comparison between modelled $Cl_y^{LL}$ abundances and MLS observations is*

*about 3.5 % and 19 % at 50 hPa for the mid-latitudes and tropics respectively, while the NME is about 2% for the global mean at 3.5 hPa (see Table S1).* [...]

**P8L7-9: "[...] introduces a continues reduction in TOC that exceeds the model ensembles variability between experiments [...]" I might have missed it, but where is the "model ensembles variability" actually quantified in the manuscript?**

We have included a figure (Fig. S2) with 9-panels in the supplementary material showing the TOC evolution for each of the three individual simulations of each experiment for the mid-latitudes and tropics. These panels include both the smoothed and non-smoothed TOC results. In addition, we show the smoothed mean-ensemble TOC result of each experiments in all the panels, in order to visually display the following points:

- there is no marked difference in the smoothed TOC variability of each simulation with respect to the smoothed ensemble-mean TOC variability throughout the modelled period, for each experiment within the mid-latitudes and tropics.

- the TOC reduction mediated by VSL-bromine exceeds the TOC variability of each simulation and therefore of the ensemble-mean TOC variability within the mid-latitudes and tropics.

P9L19-20: [...] *Equivalent results for the temporal evolution of the individual simulations of each experiment, are shown in Fig. S5.* [...]

P9L26-27: [...] *The constant emission of biogenic VSL$^{Br}$ sources, introduces a continuous reduction in TOC that exceeds the man ensemble variability between the experiments (see Fig. S5)* [...]

**P8L18: "[...] the largest model bias compared to SBUV [...]" How does the model bias compare to the uncertainty in the SBUV data? If the authors cannot quantify this it should be at least discussed here.**

First, we have included a Table (S2) in the supplemental material indicating the bias, normalized mean bias and normalized mean error as a measure to evaluate the agreement between modelled TOC and SBUV-MOD observations. In addition, we have included a general description of the SBUV- databases and their "potential" uncertainties in the methodology section (in a new paragraph), as well as a sentence in section 3.2 of the new manuscript, as shown below:

Section Methods

P6L35-P7L7: [...] *An important point to highlight about the SBUV-MOD data set (and its uncertainties), is that it was constructed from ozone profiles measured by individual SBUV instruments and retrieved using the Version 8.6 algorithm (Bhartia et al., 2013; Frith et al., 2014, 2017). To maintain consistency over the entire time series, the individual instrument records were analysed with respect to each other and absolute calibration adjustments were applied as needed based on comparison of radiance measurements during periods of instrument overlap (DeLand et al., 2012; Weber et al., 2013). During the overlap periods, the records of the involved SBUV instruments were combined using a simple average. Frith et al. (2014) have estimated the potential errors in the SBUV-MOD data set through Monte Carlo model simulations, taking into account the range of offsets and drifts observed during the overlap periods between individual SBUV instruments and between SBUV and Dobson ground-based measurements. The authors also assessed the extent to which these potential errors might affect the long-term variability of ozone in a multiple-regression model. See*

*Frith et al. (2014) for a detailed description of the data used and their associated uncertainties in the SBUV-MOD.*

Section 3.2 Results and Discussion

P9L30-32: [...] *Note is that the statistical analysis presented in Table S2 does not take into account the uncertainties that may arise from the merging of the individual SBUV instruments in the SBUV-MOD data sets (Frith et al., 2014). See Frith et al. (2014) for a detailed description of the potential associated uncertainties in the SBUV-MOD data set* (see Section 2).

**P8L20-21: "[...] much larger ozone loss efficiency [...] within the mid-latitudes [...]" Maybe point out that this is true in both hemispheres.**

We have rephrased the lines P8L20-21 of the original manuscript, as following:

P10L7-8: [...] Overall, our model results show a much higher ozone loss due to the VSL$^{Br}$ within the southern and northern mid-latitudes compared to the tropics.

**P6L32-34: "The slight underestimation [...]" If this is truly due to unaccounted anthropogenic chlorine sources, this means that the RCP 6.0 projections regarding future emissions of chlorinated species are erroneous. What are the implications on future ozone and climate? Based on their findings, the authors may discuss this in Section 4.**

**P8L38-P9L2: "[...] which shows a continuous decline in the course of the 21$^{st}$ century from its peak values observed during 2000 [...]" As mentioned above, one has to be aware at this point, that decline this is due to the emission scenario for CFCs used in the model. As mentioned above and as the authors mentioned citing Hossini (2015b, 2017), if CFCs are not out-phased as demanded by the Montreal protocol or if new anthropogenic sources of halogenated substances occur in the future the projections will change. This should be, at least, mentioned in Section 4.**

We are grateful for these last two comments of the referee regarding our evolution of inorganic chlorine (1960-2100) as well as no the potential effect that anthropogenic VSL$^{Cl}$ sources (combined with VSL$^{Br}$) could have on the evolution of stratospheric ozone during the 21$^{st}$ century. Based on his/her suggestions, we have included the following change in the Introduction and Conclusions:

Introduction:

P3L3-12: [...] *Hossaini et al. (2015a), quantified the stratospheric injection of organic and inorganic chlorine from anthropogenic VSL$^{Cl}$ sources such as chloroform (CHCl$_3$), dichloromethane (CH$_2$Cl$_2$), tetrachloroethene (C$_2$Cl$_4$), trichloroethene (C$_2$HCl$_3$), and 1,2-dichloroethane (CH$_2$ClCH$_2$Cl), which are not controlled by the Montreal Protocol. From their results, the total stratospheric chlorine load from VSL$^{Cl}$ inferred for 2013 is 123 ppt, with a stratospheric injection dominated by source gases (~ 83%) of VSL$^{Cl}$. Moreover, the stratospheric VSL-chlorine (organic and inorganic) injection increased by ~ 52% between 2005 and 2013, mainly due to a recent and ongoing growth in anthropogenic CH$_2$Cl$_2$ emissions. In fact, Hossaini et al. (2017), showed that the impact of CH$_2$Cl$_2$ on stratospheric ozone has increased markedly in recent years and if these increases continue into the future, the recovery of Earth's ozone layer could be delayed even further, offsetting some of the gains achieved by the Montreal Protocol.* [...]

Conclusions:

P15L9-14: [...] *In the same line, Hossaini et al. (2015a) demonstrated that the chlorine load in the lowermost stratosphere has increased as a consequence of a recent and ongoing growth in emissions from anthropogenic VSL$^{Cl}$ sources (mainly CH$_2$Cl$_2$) not controlled by the Montreal Protocol. If this VSL$^{Cl}$ emissions trend continues into the future, additional studies will be required to quantify the increase in VSL-bromine impact modelled in this work on the projected ozone loss trends over the century, considering the additional inorganic chlorine from VSL$^{Cl}$.* [...]

In addition, we have rephrased the lines P6L32-34, P8L38, P9L2 of the original manuscript to be more consistent with our message on this point, as shown below:

P8L7-11: [...] *For example, the normal mean error of the comparison between modelled Cl$_y^{LL}$ abundances and MLS observations is about 3.5 % and 19 % at 50 hPa for the mid-latitudes and tropics respectively, while the NME is about 2% for the global mean at 3.5 hPa (See Table S1). This relatively good agreement between the model and MLS observation occurs even without consideration of the recent contribution of anthropogenic VSL$^{Cl}$ sources (i.e., Hossaini et al., 2015b, 2017)* [...]

P10L25-31: [...] *the VSL-bromine efficiency in ozone depletion is primarily linked to the background inorganic halogen abundance, which shows a continuous decline in the course of the 21$^{st}$ century contemplated in the A1 halogen emission scenario considered in this work. This is in line with the findings of Yang et al. (2014). Therefore, an additional stratospheric chlorine and bromine load from natural or anthropogenic substances not regulated by the Montreal Protocol, will induce an increase in the VSL$^{Br}$ impacts modelled in this work on projected ozone loss trends throughout the current century.* [...]

**P11L5-P12L22: This subsection's content is rather difficult to follow at times due to the abbreviations in use. Especially the terms, Halog$_{–Loss}$, BrOx$_{–Loss}$, and VSL$^{Br}$ are confusing. There seems to be no clear seperation between in the term Halog$_{–Loss}$ and BrOx$_{–Loss}$ with and without additional Br from short–lived substances. I will come back to this later on. The authors may elaborate on their terminology in this regard.**

We have modified the opening paragraph of this section to clarify the terms used throughout this section. Also, we have decided to specify with the supra-indexes $^{LL+VSL}$ or $^{LL}$ on the different families/cycles (O$_{x-Loss}$, NO$_{x-Loss}$, HO$_{x-Loss}$, Halog$_{x-Loss}$, BrO$_{x-Loss}$, ClO$_{x-Loss}$ and BrO$_x$-ClO$_{x-Loss}$) when we refer to the simulations with and without VSL$^{Br}$ sources throughout the section.

We have modified the text as follows:

P12L35-P13L3: *In order to determine the drivers controlling the VSL$^{Br}$-driven seasonal impact on ozone in the lowermost stratosphere at mid-latitudes and tropics, we compute the odd oxygen chemical loss from the each ozone-depleting families considering the independent contributions of oxygen ($O_{x-Loss}$), hydrogen ($HO_{x-Loss}$), nitrogen ($NO_{x-Loss}$), and halogen ($Halog_{x-Loss}$) (Brasseur and Solomon, 2005; Saiz–Lopez et al., 2014). Moreover, we discriminate the individual contribution of pure bromine ($BrO_{x-Loss}$) and chlorine ($ClO_{x-Loss}$) cycles and inter-halogen ($ClO_x–BrO_{x-Loss}$) cycle to the halogen family ($Halog_{x-Loss}= BrO_{x-Loss} + ClO_{x-Loss} + ClO_x–BrO_{x-Loss}$).* [...]

**Section 4**

**In addition to the point mentioned above, since the results for the mid-latitude are substantially different from the ones for the Tropics, the authors may consider to separate these into different subsections to make it more clear.**

5    Taking into account the reviewer's general commentary, we have reordered this entire section, separating the VSL$^{Br}$-mediated impacts on the mid-latitudes and tropics. In the reorganization of this Section we have taken into account all the technical corrections indicated by the reviewer.

**1.3 Technical corrections**

10   1.3.1 General

**Although it is a matter of personal style/taste: There are many preceding conjunctions throughout the text which appear to be fillers. The authors may consider to remove those.**

As suggested by the reviewer, we have eliminated all the conjunctions throughout the manuscript that we consider necessary.

15   **The authors use º throughout the manuscript instead of °. This should be corrected for.**

We have changed "º" by "°" throughout the manuscript.

1.3.2 Specific

Abstrac

20   **L20: "[...] the inclusion of VSL$^{Br}$ result [...]" result → results**

We have changed "result" by "results"

**L33/L35 and others: "The largest modelled [...] the spring [...]" As long as not a specific spring or winter, etc. is meant, the prefix article is wrong.**

We have removed the prefix article here and in other parts of the text when we do not refer
25   to a specific season.

**L36/L37-38: "Halog$_{–Loss}$" Just a remark: In the final typesetting this ought not be broken at the page's margin.**

In the new version of the manuscript, we have made sure that all terms containing "x-Loss" are not separated in the margin of the page.

30   **L35-37: "[...] the halogen-mediated ozone depletion [...] is more efficient than [...] respect to other seasons." respect → with respect**

We have changed "respect" by "with respect"

Section 1

35   **P1L15-17: "Owing to their short lifetimes, the impact of VSL on stratospheric ozone peaks at the extratropical lowermost stratosphere [...] an important atmospheric region [...]" I suggest to insert "which is" → "[...] which is an important atmospheric region [...]" to make the sentence more readable.**

We agree with the reviewer's suggestion and have included "which is" in the sentence in
40   lines P2L18-20.

**P1L34: imply → implies**

We have changed "imply" by "implies"

**P3L2: "[...] 3,290 ppt Cl and 19.6 ppt Br [...]" Only a remark: It is rather uncommon to use ppt for such high concentrations (chlorine), but the authors probably intent to show the difference between chlorine and bromine concentrations in the most comprehensible way.**

We have changed the concentration unit for total stratospheric chlorine from ppt to ppb.

**P3L7: "[...], understanding the role of natural VSL$^{Br}$ sources is key for chemistry- climate projections." Missing article in front of "key".**

We have modified the sentence of the lines P3L7 in the original manuscript, as follows:

Thus, understanding the role of natural VSL$^{Br}$ sources is a key issue for chemistry-climate projections.

**P3L11/P4L22/P11L25: "[...] has been reviewed elsewhere [...]"/"[...] has been given elsewhere [...]"/"[...] has been described elsewhere [...]" Not sure if this is a proper citation style. Use "by" instead and have in-text citations?**

The usage of "elsewhere"-like sentences is widely extended in the literature. We have checked the citation style in ACP and there is not a specific rule about it. Thus, we rather keep the citation style of this sentences.

**P4L5-9: "The layout of the paper is as follows: Section 2 resumes the main charac- teristics [...]" I assume "resume" is not the right word here. The authors probably meant "summarize".**

We have changed "resume" by "summarize"

**Section 2**

**P4L16-17: "[...] with the exception that here we consider a constant and geo-graphically distributed and seasonally-dependent oceanic emissions [...]" Substitute "here" with "in this work" or something similar; emissions → emission.**

**P4L18-19: "It is worth noting that this emission inventory [...], which allows in- troducing a complete seasonal cycle on the emission strength [...]" "It is worth noting that" can be dropped. As mentioned above, it merely serves as filler in this context;"allows introducing [...] on" → "allows to introduce [...] to".**

We agree with the corrections and changes suggested by the reviewer in the last two comments, and have changed the lines P4L16-19 in the original manuscript, as follows:

P4L28-34: [...] *The model setup follows the Chemistry-Climate Model Initiative (CCMI) REFC2 configuration described in detail by Tilmes et al. (2016), with the exception that in this work we consider a full halogen chemistry mechanism and oceanic emission –seasonal-dependent and geographically distributed– of six bromocarbons (VSL$^{Br}$= CHBr$_3$, CH$_2$Br$_2$, CH$_2$BrCl, CHBrCl$_2$, CHBr$_2$Cl and CH$_2$IBr) from Ordóñez et al., (2012) emission inventory. This emission inventory is based on a monthly-varying satellite chl-a climatology, which*

*allows to introduce a complete seasonal cycle to the emission strength and spatial distribution of oceanic bromocarbons.* [...]

**P4L22-23: "Monthly and seasonally varying zonally averaged distributions lower boundary conditions of long-lived chlorine [...] were considered [...]" I don't un- derstand this sentence. Did the authors mean to write: "Monthly and seasonally varying zonally averaged distributions of long–lived chlorine [...] and bromine [...] were considered as lower boundary conditions [...]"?**

**P4L22-28: "Monthly and seasonally varying [...]" This sentence is way to long. I suggest to break it up into two sentences. The natural breaking point would be L26 "[...], while [...]".**

We agree with the corrections and changes suggested by the reviewer in the last two comments, and have changed the lines P4L22-28 in the original manuscript, as follows:

P4L37-P5L3: [...] *Monthly and seasonally varying lower boundary conditions were considered for long-lived chlorine ($LL^{Cl}$ = $CH_3Cl$, $CH_3CCl_3$, $CCl_4$, CFC–11, CFC–12, CFC–113, HCFC–22, CFC–114, CFC–115, HCFC–141b, HCFC–142b) and long–lived bromine ($LL^{Br}$ = $CH_3Br$, H–1301, H–1211, H–1202, and H–2402), following the A1 halogenated ozone–depleting substances emissions scenario from WMO Ozone Assessment Report (2011). The surface concentrations [...].*

**P4L28-29: "In order to avoid unnecessary uncertainties associated with the speculative evolution of $VSL^{Br}$ [...]" The terms "unnecessary" and "speculative" are too judgmental in this context. You may rephrase to "[...] reduce uncertainties associated with the uncertain evolution [...]".**

In accordance with the reviewer's commentary, we have modified the lines P4L28-31 of the original manuscript as follows:

P5L32-35: [...] *Therefore, in this work, we forced the model with annually-cycled $VSL^{Br}$ fluxes, replicating the Ordoñez et al. emissions inventory for all years between 1950 and 2100. This procedure aims to reduce the uncertainties associated with the unknown evolution of both biogeochemical production and oceanic emissions of $VSL^{Br}$ into the future (Lennartz et al.; 2015; Ziska et al., 2017).*

**P4L29-31: "[...] we used a constant annual source strength for the whole modelled period [...]" A "constant annual source strength" somehow implies constant emissions which is probably not what the authors intent to say based on the previous sentences and the half-sentence in parenthesis which follows. It is clear that they treat the Ordoñez et al. emissions inventory as climatology, not projecting any future increase/decrease in emissions or changes in the seasonal cycle which they call "speculative". The authors should make this more clear.**

We appreciate this comment from the reviewer, as it has allowed us to make this point clearer.

We agree with the reviewer that the phrase "constant annual source strength" somehow implies constant emissions, which is not what we mean. We treat the Ordoñez et al. inventory as climatology, without projecting any future increase/decrease year after year, but maintaining the changes in the seasonal cycle.

To make these points clear, we removed lines P4L28-31 of the original manuscript and included the following lines in a new paragraph in Section 2:

P5L32-33: [...] *Therefore, in this work, we forced the model with annually-cycled VSL$^{Br}$ fluxes, replicating the Ordoñez et al. emissions inventory for all years between 1950 and 2100.* [...]

**P4L32: "The CAM-Chem configuration used here [...]" here → in this work.**

We have changed "here" by "in this work"

**P5L21-22: "[...], the zonal mean of the ensemble mean of each independent experiment (run$^{LL}$ and run$^{LL+VSL}$) [...]" The usage of the term "run" seems to be ambiguous throughout the text. On P5L10/L13, a run refers to one single model integration or set-up but in this sentence and later on it seems to refer to the ensemble mean? The authors should elaborate on this.**

We used the run$^{VSL+LL}$ and run$^{LL}$ terms of the original manuscript, to differentiate the results of experiments that consider and neglect the VSL$^{Br}$ sources. However, as the reviewer suggests, we have removed the run$^{VSL+LL}$ and run$^{LL}$ terms throughout the work and replaced them with the following:

P6L6-7: [...] *The mean ensemble of each experiment (i.e. with and without the VSL$^{Br}$ sources) was calculated as the average of the three individual simulations.* [...]

**P5L33-37: Maybe the use of a list could increase the readability of this paragraph?**

As suggested by the reviewer, we have rephrased the last part of the paragraph into a list to increase the readability of this paragraph.

**Section 3**

**P6L5: "In addition, Figure 1 also [...]" Either "in addition" or "also" is unnecessary in this sentence, because they mean the same in this context. Please drop one or the other.**

We have eliminated the "also" word

**P6L7: "[...] the temporal evolution [...] show [...]" show → shows**

We have changed "show" by "shows"

**P6L14/L46: present-time → present–day; present times → present–day**

We have changed "present-time" by "present-day"

**P6L19-22: "Although within the tropical lower stratosphere [...], the contribution [...]" Consider to split this sentence at "the contribution".**

We have rephrased the lines P6L19-22 in the original manuscript, as following:

P7L28-31: [...] *Furthermore, the contribution from VSL$^{Br}$ to mid-latitude total inorganic bromine also reaches ~5 ppt throughout the modelled period, while in the tropics, prevails a small carbon-bonded organic fraction that has not been converted to the reactive inorganic form.* [...]

**P6L34: "Moreover, although [...]" Filler, conjunctions can be removed.**

We have modified the beginning of the sentence in the lines P6L34 of the original manuscript, as follows:

P8L13-15: [...] *Note that although in the Section 3.4 we processed output at 120 hPa in the mid-latitude lowermost stratosphere, we did not compare $Cl_y^{LL}$ at this altitude level* [...]

**P7L18-20: "A clear example is the fingerprints [...]" This sentence is not clear. Do the authors mean to say: That the fingerprints of the Indian summer monsoon can be seen in the ClOx/Cly ratio at the given location (15 °N and 100 hPa)?; is → are**

We have modified the sentence in the lines P7L18-20 of the original manuscript, as follows:

P8L36-38: [...] *In fact, the experiments capture a local increase in the $ClO_x/Cl_y$ ratio at 15ºN and ~100 hPa, as a result of a large increase in the $Cl_y$ species heterogeneous reactivation due to the enhanced vertical transport during the Indian summer monsoon (Solomon et al., 2016b)* [...]

**P8L5-6: "A list of interesting features can be observed [...]" This phrase sounds colloquial. Maybe change to "Many interesting features [...]".Talking about lists, the author may consider to actually use a list for their points (i)-(iv). This would increase the readability of the text and content at this point.**

We agree with the reviewer's suggestion, and have rephrased and incorporated a list of our highlighted points in the lines P8L5-6 of the original manuscript, as follows:

P9L24-25: [...] *Based on the comparison of the TOC results over the different latitudinal bands, the following features can be described:* [...]

**P8L33: "[. . . ] the seasonal relatives [...]" It is not clear, what the authors intent to say here, especially since "relatives" (= family members) is the wrong word. Do they mean the "relative difference in ∆TOC for each season"? This should be clarified.**

**P8L34: "dotted" This is probably not the right term in this context. There are actually no "dotted" lines in Fig. 5. All lines are solid, but some (in the lower part of the plot) display gaps which indicate the significance of the changes. The authors may consider changing their wording here.**

We have changed the sentence in the lines P8L33-34 of the original manuscript, taking into account the last two comments of reviewer, as follows:

P10L19-21: [...] *Moreover, note that when comparing the seasonal relative ∆TOC between both periods, a statistically significant difference (here defined as p < 0.05) is only observed at the mid-latitudes, as shown by the horizontal lines at the bottom of Fig. 5d.*

**P9L16-21: "In contrast to mid-latitudes [...] for the tropics [...]" The results for the Tropics are the total opposite compared to mid–latitudes. To not confuse the results, it would increase the readability if this paragraph was to begin in a new line.**

We agree with the reviewer's comment, and have separated the results of tropics with respect to mid-latitude into a new paragraph as follows:

P11L7-10: *Within the tropics, no significant changes in the TOC loss due to VSL$^{Br}$ is projected, though the model captures a slightly major TOC depletion during the 1990s.* [...]

**P9L32-33: "In contrast [...] is at least half-fold [...]" "half-folded" is most likely not the proper term in this context. Did the authors mean that ozone loss due to VSL$^{Br}$ is reduced by 50 % by the end of the 21st century compared to present–day?**

As the reviewer notes, we have tried to say that the VSL$^{Br}$-mediated ozone loss is reduced by 50% by the end of the 21$^{st}$ century compared to present–day.

We have rephrased the lines P9L32-33 of the original manuscript:

P11L23-24: [...] *In addition, the O$_3$(z) reduction due to VSL$^{Br}$ in the mid-latitudes lowermost stratosphere is* about 50% smaller *by the end of the 21$^{st}$ century period compared to the present-day period.* [...]

**P9L34: "In agreement with our results, [...]" This should be rephrased using "our results are in agreement with".**

We have rephrased "In agreement with our results" by "Our results are in agreement with"

**P9L37: "Interestingly, deepest O$_3$(z) reductions [...]" Drop "interestingly" if you don't mean to say that this is completely unexpected; here and similar at other places (e.g. P10L16): deepest → largest / greatest.**

We have eliminated the word "interestingly" and changed the word "deepest" by "largest"

**P10L24-27: Some missing white spaces in front of or behind "-"; "Montreal Protoco" missing "l"**

We have corrected the word "Protocol"

**P10L28: "This is line [...]" → "This is in line [...]"**

We have changed "This is line" by "This is in line"

**P11L13: "[...] over the SG-ML lowermost stratosphere [...]" over → in**

We have changed "over the SH-ML lowermost stratosphere" by "in the SH-ML lowermost stratosphere"

**P11L30: seasonality increase → seasonal increase**

We have changed "seasonality increase" by "seasonal increase"

**P11L37: "crossed ClO$_x$−BrO$_{x–Loss}$" Do you mean "mixed"?**

We have changed the term "crossed BrO$_{x–Loss}$" by "inter-halogen ClO$_x$−BrO$_{x–Loss}$"

**Section 4**

**P13L18: "[...] winter. with maximum [...] reaching" with → The; reaching → reaches**

We have changed ".with" by "The"; and "reaching" by "reaches"

**P13L19-21: "We find that the inclusion of VSL$^{Br}$ leads to seasonal changes on the overall depth and vertical distribution of ozone [...], which could result to [...]" This sentence is not clear. By "overall depth", do you mean the TOC?**

We have rephrased the sentence of the lines P13L19-21 of the original manuscript, as follows:

P15L16-19: [...] *We find that the inclusion of VSL$^{Br}$ leads to seasonal changes in both local O$_3$(z) reduction and its vertical ΔO$_3$(z) distribution within the lowermost stratosphere, which could result in different seasonal perturbations on the radiative effects mediated by ozone.*

**changes on the → changes in the; results to → results in**

We have changed "changes on the" by "changes in the"; and "results to" by "results in"

**P13L30-33: "[...] entirely dominated by BrO$_{x-Loss}$ [...]" Also in Section 3 and elsewhere in the manuscript, the authors use the term BrO$_{x-Loss}$ to indistinguished between ozone depleted by BrOx from long–lived opposed BrOx from short–lived substances (VSL$^{Br}$)? This is somewhat confusing, as already mentioned above, since all organic bromine has to be transformed into inorganic bromine to deplete ozone.**

As mentioned in a previous comment, taking the reviewer's suggestion, we have specified with the sub-indexes $^{LL+VSL}$ or $^{LL}$ on the different families and/or cycles when we refer to the simulations with and without VSL$^{Br}$ sources throughout sections 3 and 4. For example, we differentiate the contribution of BrO$_{x-Loss}$ cycles to Halog$_{x-Loss}$ with the terms BrO$_{x-Loss}^{LL+VSL}$ and BrO$_{x-Loss}^{LL}$ for the simulations with and without VSL$^{Br}$.

**Figures:**

**The authors should, in general elaborate on:**

– **be more concise in captions. It is not always easy to see (without reading the whole caption) what each panel actually refers to since the relevant description is embedded in long sentences.**

– **choose a different color map for most of their contour plots since the "rainbow colors" imply visually a distinct divergence of data at the edge between blue and green/yellow. This may lead to unintended misinterpretation[1]. It is therefore depreciated, but unfortunately still widely used.**

We have a written a concise caption for all figures and have changed the "rainbow" colour palette in all graphics to a "white-red" colour palette.

**Fig. 1**

**With respect to the most common type of color-blindness, the authors may refrain from using green together with red in plots when there is no difference in line style or luminescence of the colors.**

**Caption**

**L3-4: "(a) global upper stratosphere (6.5 hPa) and lower stratosphere (50 hPa) (b) tropics [...]" The way it is written, it looks as if "lower stratosphere [...]" is part of panel (a) instead of (b)-(d). The authors should make this more clear. Maybe use ";" instead of "and".**

**L5: "show changes" Wouldn't it be use the term "evolution" for consistency?**

**L7: "The red triangles of all panels [...]" You can drop "of all panels".**

**L9: dotted → dashed**

We have changed the original green colour to a light blue in all the plots and modified the sentences of the Fig.1 caption, to increase its readability, as follows:

Temporal evolution (1960–2100) of the modelled annual mean abundances of $VSL^{Br}$ and inorganic halogen ($Cl_y^{LL}$ and $Br_y^{LL+VSL}$): (a) global upper stratosphere (3.5 hPa); lower stratosphere (50 hPa) at (b) tropics, (c) SH-ML and (d) NH-ML. [...] The black squares (filled and open) and orange triangles (filled and open) of the panel (a) show the total $Br_y$ evolution (1991-2012) reported in the latest WMO (2014; 2018) [...]

We have changed "dotted" by "dashed"

We have eliminated the term "of all panels"

**Fig. 2**

**See comment about color map.**

**Fig. 3**

**Panel labels (a)-(i) move around quite a bit. Would it be possible to set them to lower left corner for each plot?**

We have modified the figure to unify the panel label positions.

**Fig. 6**

**The y-axes' ranges changes between the plots ((a,c) 300–4; (b) 100–4; (d,e,f) 300–5) hPa. Please make the range consistent in all of the plots.**

**Panel (f): See above comment on color map. Caption**

**L2: within of → within the**

**L3: shows → show Fig. 8**

We have unified the 300-5 hPa scale for all the plots. We have also changed "within of" by "within the"; and "shows" by "show"

**Fig. 8**

**The axes' lines are far thicker than the actual markers or lines symbolizing the data which is distracting. Partly due to this the violet line and especially the black line at the very bottom in panel (a) are easily missed. Please reduce the axes thickness and think about to change the color for Ox−Loss.**

**The legends are overlapping with the content and should be placed outside of the plot area (right hand side).**

**The plots (b) and (c) are extremely busy and not well explained in the caption. Why are data for JJA displayed using dashed lines in contrast to the thin, solid line/marker combination used for the remaining "seasons"? You should rather use either four different line styles or markers. Please make this consistent.**

**Is it possible to show the actual ensemble standard deviation in this plot? It is mentioned in the text but nowhere quantified.**

**Caption**

**L4-6: "The panels (b) and (c) show the seasonal mean contributions [...]" This is not concise. Already is it difficult to spot LL and LL+VSL in the plots' legends, but the way this is presented in the caption is not making it any easier to interpret this plot. The actual information in what (b) and (c) differ comes in the very end of a long sentence. A better approach would be, e.g. "Seasonal mean contributions of [...] families for the different experiments (b) run$^{LL+VLS}$ and (c) run$^{LL}$."**

We have made the following changes to this figure and the corresponding ones shown in the supplementary material:

- reduced the markers on all the plots

- adjusted the x-axis range of all the plots (a)

- changed the colour for the $O_{x\text{-Loss}}$ family

- changed the style/marker for JJA data in the panels (b) and (c)

- moved the legends to the right margin of each panel.

We consider that the changes made to the figure format will help to better understand the description of the captions.

Perhaps there was a phrase in section 3.4 of the original manuscript that led the reviewer to think that we computed the standard deviation of the families for each experiment. We have controlled the wording of the entire section, to avoid confusion about this in the new version of the manuscript. However, we consider that it is not necessary to show the standard deviation in this type of plot, since the seasonal changes of the families in the individual simulations for each experiment are very similar and therefore there is almost no deviation of these from ensemble mean of each experiment.

**Fig. 9**

**See comment about color map above.**

**The color bar indicates an open interval in both direction (blue and red tri- angles). If the data actually is confined in a closed interval between 0 and 100, the triangles should not to be used.**

**Can you show or quantify the magnitude of difference between the percental contribution of BrO$_{x–Loss}$, BrO$_x$-ClO$_{x–Loss}$, and ClO$_{x–Loss}$ in both, run$^{LL}$ and run$^{LL+VSL}$? One cannot draw this information of a plot with color levels of widths 10 % especially for panels (c,d). (Even worse in Fig. S9).**

We appreciate the reviewer's comments on these types of figures. We have changed the colour map in the 0-100% range, with a 5% interval. On top of the colour maps we have superimposed line contours to help visualize the changes in the contribution of each cycle to the Halog$_{x\text{-Loss}}$ family, as discussed in section 3.4. We consider that the combination of color and line contours also helps to quantify the contribution changes between the cycles of the

same experiment, as well as the contribution changes of the same cycle between the experiments.
* * *
5 **2   Anonymous Reviewer #2**
* * *
**The manuscript presents a modelling study of the impact of very short-lived halogenated compounds (VSL$^{Br}$) on stratospheric ozone under current and future conditions. In particular, seasonal changes of the impact VSL$^{Br}$ on midlatitude ozone and their link to**
10 **heterogeneous chlorine reactivation are discussed. While many findings of this study are known from existing publications, the seasonality of the changes presented here is a unique and new result. The manuscript is well written, and the study is of interest to the readership of ACP. I recommend publication after addressing the following comments.**

15 **2.1   General comments**

**1) The CAM-Chem version used here does not include the full stratosphere and has a relatively low lid when compared with state-of-the-art chemistry climate models. Therefore, the full Brewer-Dobson circulation path is not included in the model domain. It is not clear how modelled future changes of the circulation and stratospheric**
20 **temperatures are impacted by the low model lid and a discussion of these facts should be included in the manuscript.**

Our full VSL$^{Br}$ state-of-the-art chemical scheme has only been implemented in CAM-Chem, and not in higher top models such as WACCM. Thus, we cannot perform this type of evaluation using a high-top model.

25 In particular, the region of maximum gravity-wave dragging, as well as most of the Brewer-Dobson circulation, is located above CAM-Chem model top. Therefore to reproduce the overall stratospheric circulation, the integrated momentum that would have been deposited above the model top is parameterized and deposited in the top layer of the model as an upper boundary condition (Lamarque et al., 2008), resulting in a consistent description of the
30 stratospheric dynamics.

Even when CAM-Chem has a relatively low model top (~40 km), the stratospheric performance of CAM-Chem has been extensively inter-compared against the widely used WACCM configuration of CESM (i.e., considering 56 levels up to ~80 km), showing an excellent agreement in reproducing the large-scale changes of the stratospheric chemical
35 composition, radiation and climate. For example: Fig. 2 in Lamarque et al. (2008), shows an inter-comparison of photolysis rates of active radiative species throughout the whole vertical profile, while Fig. S1 in Fernandez et al. (2017) compares the climatic evolution of the stratospheric Age of Air and Total Ozone Column evolution through the 21st century. It should be noted that CAM-Chem was part of CCMVal-2 (Eyring et el., 2010) and CMIP5
40 (Eyring et al., 2013) inter-comparison projects, presenting ozone evolution trends and return dates estimations that lies in the middle of the multi-model range. To make this clear, we have included the following lines in the "Methods" Section of the new version of the manuscript:

P5L11-16: [...] *To ensure a consistent dynamical description of the stratosphere within the*
45 *relatively low model top of CAM-Chem (i.e., gravity-wave dragging and the Brewer-Dobson circulation), the integrated momentum that would have been deposited above the model top*

*is specified by an upper boundary condition (Lamarque et al., 2008). A similar procedure is applied to the altitude-dependent photolysis rate computations, which include an upper boundary condition that considers the ozone column fraction prevailing above the model top.* [...]

5    P5L21-24: [...] *Equivalent CAM-Chem configurations, as the one implemented in this work, have been used used for the CCMVal-2 (Chemistry-Climate Model Validation 2) and CMIP5 (Coupled Model Intercomparison Project Phase 5) activities, in order to represent trends in the ozone evolution and to estimate the return dates, which lie in the middle of the multi-model range (Eyring et al., 2010; 2013)* [...]

**2) A critical discussion of the how the projected changes of lowermost stratospheric temperatures impact the role of VSL$^{Br}$ driven ozone loss is needed. Both points (1 and 2) should be used to demonstrate that the model version used here offers an appropriate model set up.**

15    The stratospheric ozone evolution in the 21$^{st}$ century will be controlled not only by the decline of ODSs but also, to a large extent, by changes in GHG concentrations, particularly $CO_2$, $CH_4$ and $N_2O$ (WMO, 2018 Chapter 3). These GHGs affect stratospheric ozone through temperature and subsequent changes in dynamics and transport. For example, the change in the GHG concentration will lead to a cooling of the upper stratosphere driving an increase

20    in the ozone budget due to the slowing down the gas-phase ozone loss reactions. Lower stratospheric ozone at middle to high latitudes may either increase or decrease moving into the future, depending on the hemisphere and on the GHG scenario (WMO, 2018 Chapter 3). Additionally, the lowermost stratospheric temperature and ozone response to varying GHGs and ODSs in different simulations finds varying degrees of disagreement between CCMI-1

25    models. Thus, to avoid introducing additional uncertainties regarding the additional ozone depletion from VSL$^{Br}$, in this work we used a single set of ODSs (A1) and GHG (RCP 6.0) scenarios. From the analysis of the two CAM-Chem simulations, we conclude that the particular impact of the VSL$^{Br}$ on the lowermost stratospheric ozone budget depends on a seasonal interplay between the heterogeneous reactivation of inorganic halogens ($Br_y$ and

30    $Cl_y$) and the decrease in the abundance of long-lived background halogens.

In relation to the version and configuration of the CAM-Chem model:

The combined (online + lookup table) photolysis approach used in CAM-Chem -corrected in the upper boundary by including an additional layer of ozone and oxygen to represent the stratospheric fraction remaining above the model top- provides a very accurate

35    representation of photolysis rates at all altitudes, including the lowermost stratosphere. Indeed, the CAM-Chem altitude distribution of ozone abundances, as well as the of O(1D) and O(3P) photolytic production, are in excellent agreement when compared to an equivalent simulation performed using a high-top stratospheric model (WACCM, see Fig. 2 in Lamarque et al., 2008). Moreover, the fully resolved stratospheric module was evaluated in

40    Chapter 6 of the SPARC CCMVal report, where CAM-Chem was shown to possess one of the more accurate photochemical modules used in CCMs (see SPARC, 2010 for details). As a conclusion of the CCMVal-2 intercomparison project, it was mentioned that the temperature evolution in the mid-latitude lower stratosphere for the multimodel mean is similar to that in the tropics, though the temperature responds more to changes in ODSs than

45    to GHGs, in particular in the southern midlatitudes (Eyring et al., 2010). Given that CAM-Chem uses identical ODS LBC than high model tops such as WACCM, we consider that the model presents a reliable representation of lowermost stratospheric temperatures.

In addition, we repeat below the answer to reviewer No 1, related to this issue.

The interactive ocean coupling is recommended for climatic REFC2 1950-2100 simulations to make the representation of climate change in the models more physically self-consistent. To isolate the "chemical impact" of VSL[Br] in the present time atmosphere, we've already performed perpetual 2000 sensitivity simulations (with/without VSL sources) forced with identical meteorological fields, and determine the changes in inorganic bromine burden, abundance and impacts (Fernandez et al., 2014). However, to determine the climatic impact of VSL[Br] sources into an evolving stratosphere, we need to explicitly consider the radiative and temperature feedbacks. Indeed, the REFC2 CCMI recommendation suggests performing coupled sea-ice and ocean simulations for the 1950-2100 periods (see Table 2 in Eyring et al., 2013, where it is explicitly mentioned that "*Development should continue towards comprehensive troposphere-stratosphere CCMs, which include an interactive ocean, tropospheric chemistry, a naturally occurring QBO, spectrally resolved solar irradiance, and a fully resolved stratosphere (P49)*"). We truly believe that the current CAM-Chem REFC2 configuration considers all these issues and have been already widely validated in both the troposphere and stratosphere (see previous answers). To highlight all of this, we have incorporated the following sentences in the corrected manuscript:

P5L25-28: [...] *This implies that instead of isolating the chemical impact mediated by VSL[Br] sources on the climatological ozone budget (as would be the case using a specified dynamic approach), the chemical interaction between VSL-bromine and ozone in the lowermost stratosphere is affected by dynamic feedbacks (i.e. temperature, radiation, etc.).* [...]

**3) The discussion of the VSL[Br] emission parametrization should make clear that the seasonality of the emissions is not well known due to missing process understanding and sparse observational data.**

Based on the reviewer's comment, we have included the following lines in a new paragraph in the "Methods" Section, to make this point clear:

P5L29-35: [...] *Even though the interactive ocean coupling would have allowed us to compute the evolution of VSL[Br] ocean source emissions throughout the modelled period, we still lack a considerable understanding of the seasonal processes that dominate VSL[Br] ocean emissions in both present and future time, which is combined with a limited observation data set on VSL bromocarbons (WMO, 2014; WMO 2018). Therefore, in this work, we forced the model with annually-cycled VSL[Br] fluxes, replicating the Ordoñez et al. emissions inventory for all years between 1950 and 2100. This procedure aims to reduce the uncertainties associated with the uncertain evolution of both biogeochemical production and oceanic emissions of VSL[Br] into the future (Lennartz et al.; 2015; Ziska et al., 2017).*

**4) Why are chlorinated very short-lived substances not included in the halogen budget? Their current and potential future emissions could play an important role and it would be interesting to see how this impacts the relative role of VSL[Br].**

We agree with the reviewer that considering anthropogenic VSL[Cl] sources (and their potential future emissions) would increase the ozone loss and the relative role of VSL[Br] sources into the future. Unfortunately, we do not currently have a VSL[Cl] inventory implemented and validated in the CAM-Chem model. This is currently under development in our group, with collaborations from other scientific groups.

To make this important point clear, we have included a general introduction on the continuous and recent anthropogenic chlorine loading from VSL sources in the "Introduction" of the new manuscript version, as well as a brief discussion in the "conclusion" on the effect that this stratospheric chlorine loading would have on our impacts modelled by VSL[Br] sources:

Introduction:

P3L3-12: [...] *Hossaini et al. (2015a), quantified the stratospheric injection of organic and inorganic chlorine from anthropogenic $VSL^{Cl}$ sources such as chloroform ($CHCl_3$), dichloromethane ($CH_2Cl_2$), tetrachloroethene ($C_2Cl_4$), trichloroethene ($C_2HCl_3$), and 1,2-dichloroethane ($CH_2ClCH_2Cl$), which are not controlled by the Montreal Protocol. From their results, the total stratospheric chlorine load from $VSL^{Cl}$ inferred for 2013 is 123 ppt, with a stratospheric injection dominated by source gases (~ 83%) of $VSL^{Cl}$. Moreover, the stratospheric VSL-chlorine (organic and inorganic) injection increased by ~ 52% between 2005 and 2013, mainly due to a recent and ongoing growth in anthropogenic $CH_2Cl_2$ emissions. In fact, Hossaini et al. (2017), showed that the impact of $CH_2Cl_2$ on stratospheric ozone has increased markedly in recent years and if these increases continue into the future, the recovery of Earth's ozone layer could be delayed even further, offsetting some of the gains achieved by the Montreal Protocol.* [...]

Conclusions:

P15L9-14: [...] *In the same line, Hossaini et al. (2015a) demonstrated that the chlorine load in the lowermost stratosphere has increased as a consequence of a recent and ongoing growth in emissions from anthropogenic $VSL^{Cl}$ sources (mainly $CH_2Cl_2$) not controlled by the Montreal Protocol. If this $VSL^{Cl}$ emissions trend continues into the future, additional studies will be required to quantify the increase in VSL-bromine impact modelled in this work on the projected ozone loss trends over the century, considering the additional inorganic chlorine from $VSL^{Cl}$.* [...]

**5) Please explain why the polar latitudes are not analysed.**

A recent study by our research group evaluated the $VSL^{Br}$ impact on the evolution of Antarctic ozone hole using the same set of simulations used here (Fernandez et al., 2017), so we decided to constrain our seasonal study to the tropics and mid-latitudes. In addition, as indicated by the reviewer and as far as we know, there are no previous studies evaluating the seasonal impacts mediated by $VSL^{Br}$ on the stratospheric ozone in the mid-latitudes and tropics (60º N to 60º S), so we decided to focus all our analysis on these regions.

**2.2 Specific comments**

**1) Page 5, line 6-7. Doe this refer to ocean dynamics or also to ocean biogeochemistry?**

We refer to the atmospheric coupling with ice/ocean dynamics. Neale et al. (2013) describe in detail the different components of the CESM1 (Community Earth System Model Version 1), including the POP2 (Parallel Ocean Program version 2) and the Los Alamos sea ice model. We have included the corresponding reference:

P5L24-25: [...] *Moreover, our model configuration uses a fully coupled Earth System Model approach, i.e. the ocean and sea ice are explicitly computed (Neale et al., 2013).* [...]

**2) Figure 1. Legend is too small, and therefore it was very hard to read the figure.**

In this new version of the manuscript, we have worked on all the details of the figures and their captions to improve the quality and description of all the figures. In particular, in Fig. 1, we took the common legends from all the panels, and inserted them in-between the panels.

**3) Page 6, line 32-33. It is not clear why not including the short-lived chlorinated species would cause an offset to the observed Cly only in the tropics. Once they contributed to the inorganic chlorine budget, they should play a role at all latitudes.**

We have rephrased the lines P6L32-33 of the original manuscript and included a new Table S1 in the supplementary material to make this clear.

P7L13-15: [...] *The values of bias, normalized mean bias (NMB) and normalized mean error (NME) to evaluate the agreement between the modelled $Cl_y$ and observations for each of the regions under study, are shown in Table S1.* [...]

P8L5-10: [...] *The modelled inorganic chlorine abundance shows a good agreement with the (HCl + ClO) MLS observations for the 2005-2015 period, mainly in the upper stratosphere where most chlorine has already been photochemically converted to $Cl_y^{LL}$. For example, the normal mean error of the comparison between modelled $Cl_y^{LL}$ abundances and MLS observations is about 3.5 % and 19 % at 50 hPa for the mid-latitudes and tropics respectively, while the NME is about 2% for the global mean at 3.5 hPa (see Table S1). This relatively good agreement between the model and MLS observation occurs even without consideration of the recent contribution of anthropogenic $VSL^{Cl}$ sources (i.e., Hossaini et al., 2015b, 2017)* [...]

**4) Page 8, line 13. Here and in several other places, a consistency between different values is cited that is not necessarily clear to the reader. Why is the number of the $VSL^{Br}$ impact being 50% at the end of the century consistent with projected delta TOC changes of 10 to 15%?**

We have computed the TOC trends between preset-day and the end of the 21$^{st}$ century, taking into account the percentage change of TOC per decade (i.e. ΔTOC trends  % dec$^{-1}$). For example, as detailed in Table 1, the difference in TOC driven by the $VSL^{Br}$ at the end of the century ($\Delta TOC^{2100}$= −1.2 %) at SH-ML is "approximately 50% lower" than the corresponding value for present-day ($\Delta TOC^{2000}$= −2.5 %). This results in a positive ΔTOC trend per decade of 0.15 % dec$^{-1}$, indicating a smaller $VSL^{Br}$ impact on TOC towards the future even if the stratospheric injection of $VSL^{Br}$ and $Br_y^{VSL}$ remains constant over the course of the current century. Note that in the original manuscript we did not suggest that the ΔTOC changed from 10 % to 15% in any of the regions analyzed.

For ozone trends in the vertical distribution of ozone, these $\Delta O_3(z)$ trends were computed for each pressure level between the present day period and the end of the 21$^{st}$ century.

P9L36-38: [...] *Moreover, $VSL^{Br}$ impacts on TOC by the end of the 21$^{st}$ century are ∼ 50% lower than the values found for the present-day, which is in line with the projected ΔTOC trends of 0.15 and 0.10 % dec$^{-1}$ for the SH-ML and NH-ML, respectively.* [...]

**5) Page 8, line 19. Why is the minimum TOC shifted to 5 years earlier when including a constant $VSL^{Br}$ term?**

From the modelled impacts on lowermost stratospheric ozone and TOC throughout the modelled period, we conclude that the $VSL^{Br}$ impact on ozone is less sensitive to abundances of background halogens in the tropics compared to mid-latitudes. However the increase of stratospheric inorganic bromine due to $VSL^{Br}$, leads to both a $TOC^{LL+VSL}$ depletion and a shift of the minimum $TOC^{LL+VSL}$ towards where the inorganic chlorine peak is located (see Figure 1). This minimum $TOC^{LL+VSL}$ shift responds to a improvement in ozone loss driven by the inter-halogen $ClO_x$-$BrO_{x-Loss}$ cycle when including the $VSL^{Br}$ sources.

We have rephrased the lines P8L19 of the original manuscript to make this clearer, as follows:

P10L4-6: [...] *Moreover, even though the $VSL^{Br}$ slightly worsens the agreement between the modelled $TOC^{LL}$ and observations, the minimum $TOC^{LL+VSL}$ is shifted ~5 years earlier compared to $TOC^{LL}$ (i.e. towards where the inorganic chlorine peak is located in year 1995), in agreement with observations*.

**6) Page 8, line 34 and other places. What is the term $VSL^{Br}$ driven ozone loss efficiency referring to? Is this describing the total impact of $VSL^{Br}$ on ozone or some relative terms (% ozone loss)?**

We use this term to refer to the total (absolute and relative) impact on stratospheric ozone and TOC mediated by the additional bromine from $VSL^{Br}$ sources. When we refer to the absolute or relative term in particular, we explicitly mention it as for example:

P9L19 [...] absolute and relative TOC differences ($\Delta$TOC) [...]; P12L5 [...] springtime absolute $O_3(z)$ reductions [...]; P12L21 [...] relative $\Delta O_3(z)$ [...]; from original manuscript.

We appreciate the reviewer's question, as this has persuaded us to check throughout Sections 3.2 and 3.3 that this term refers only to the total impact.

**7) Page 9, line 1. Something is missing in the sentence.**

We have rephrased the lines P9L1 of the original manuscript to make this point clearer, as follows:

P10L25-31: [...] *the VSL-bromine efficiency in ozone depletion is primarily linked to the background inorganic halogen abundance, which shows a continuous decline in the course of the 21$^{st}$ century contemplated in the A1 halogen emission scenario considered in this work. This is in line with the findings of Yang et al. (2014). Therefore, an additional stratospheric chlorine and bromine load from natural or anthropogenic substances not regulated by the Montreal Protocol, will induce an increase in the $VSL^{Br}$ impacts modelled in this work on projected ozone loss trends throughout the current century*.

**8) Page 11, line 20 and 21. This is not clear and might need some more explanation. First of all, Figure 8 (which has again a way too small legend) seems to suggest that the Ox and NO$_x$ terms also increase. Is this increase maybe similar to the one of the HOx term relative to the total contribution? If not, it would be worthwhile explaining why the VSL impact gets compensated by the HO$_x$ term alone.**

We have made the following changes to this figure and the corresponding ones shown in the supplementary material:

- reduced the markers on all the plots
- adjusted the x-axis range of all the plots (a)
- changed the colour for the $O_{x\text{-}Loss}$ family
- changed the style/marker for JJA data in the panels (b) and (c)
- moved the legends to the right margin of each panel.

We consider that the changes made to the figure format will help to better understand the description of the captions.

We have made an additional Figure (A1) in which we show the vertical profile of all families that contribute to ozone loss rate for year 1995 at NH-ML, tropics and SH-ML. This figure includes the results of the experiments with and without the VSL$^{Br}$ sources. The horizontal black dotted lines on each panel highlight the altitude level at which the temporal evolution (1960-2100) of each family is shown in the Figs 8a, S7a and S8a.

As shown in Fig. A1, the ozone loss rate is exclusively dominated by the $HO_{x-Loss}$ family at 50 hPa in the tropics and 120 hPa in the mid-latitudes (in both SH-ML and NH-ML). In line, $Halog_{x-Loss}$ represents the second contribution at the mid-latitudes, while within the tropics, it represents the third contribution after $NO_{x-Loss}$. Note that this "order" in the contribution of the families $Halog_{x-Loss}$, $NO_{x-Loss}$ and $O_{x-Loss}$, changes slightly during the modelled period at NH-ML and tropics, regardless of the role that VSL$^{Br}$ play, as seen in the dotted lines in Figs. S7a and S8a.

Therefore, considering the dominant $HO_{x-Loss}$ contribution in the lowermost stratosphere at mid-latitudes (120hPa) and tropics (50 hPa), the enhancement in the $Halog_{x-Loss}^{LL+VSL}$ contribution with respect to $Halog_{x-Loss}^{LL}$ by including VSL is mostly compensated by a decrease in $HO_{x-Loss}^{LL+VSL}$. This is clearly reflected in the SH-ML lowermost stratosphere - as indicated in the original manuscript - where the $Halog_{x-Loss}$ contribution represents the second contribution throughout the modelled period after the dominant $HO_{x-Loss}$.

P13L9-14: [...] *$Halog_{-Loss}$ represents the second most important contribution to total ozone loss rate after $HO_{x-Loss}$  (~ 80 %) at the mid-latitude in both experiment (see Figs. 8a and S7a), while within the tropics it represents the third family after $HO_{x-Loss}$ and $NO_{x-Loss}$ (see Fig. S8a). This partially explains the smaller modelled VSL-bromine impact on ozone within the tropics, as $Halog_{x-Loss}^{LL+VSL}$ represents at most 10% of the total ozone destruction over this region. Moreover, the enhancement in the $Halog_{x-Loss}^{LL+VSL}$ contribution with respect to $Halog_{x-Loss}^{LL}$ by including VSL$^{Br}$, is mostly compensated by a decrease in the $HO_{x-Loss}^{LL+VSL}$ contribution within the mid-latitudes.* [...]

[Figure]

Figure A1: 1995-mean vertical profiles of the contributions (%) of each family to the ozone loss rate for experiments with (ens$^{LL+VSL}$)and without (ens$^{LL}$) VSL$^{Br}$ at the northern (NH-ML, a) and southern (SH-ML, c) mid-latitudes and tropics (b). Tropical tropopause layer (TTL); upper troposphere-lower stratosphere (UTLS).